# The interdependence of excitation and inhibition for the control of dynamic breathing rhythms

Nathan Andrew Baertsch [1], Hans Christopher Baertsch[1] & Jan Marino Ramirez[1,2,3]

The preBötzinger Complex (preBötC), a medullary network critical for breathing, relies on excitatory interneurons to generate the inspiratory rhythm. Yet, half of preBötC neurons are inhibitory, and the role of inhibition in rhythmogenesis remains controversial. Using optogenetics and electrophysiology in vitro and in vivo, we demonstrate that the intrinsic excitability of excitatory neurons is reduced following large depolarizing inspiratory bursts. This refractory period limits the preBötC to very slow breathing frequencies. Inhibition integrated within the network is required to prevent overexcitation of preBötC neurons, thereby regulating the refractory period and allowing rapid breathing. In vivo, sensory feedback inhibition also regulates the refractory period, and in slowly breathing mice with sensory feedback removed, activity of inhibitory, but not excitatory, neurons restores breathing to physiological frequencies. We conclude that excitation and inhibition are interdependent for the breathing rhythm, because inhibition permits physiological preBötC bursting by controlling refractory properties of excitatory neurons.

[1] Center for Integrative Brain Research, Seattle Children's Research Institute, 1900 9th Avenue JMB10, Seattle, WA 98101, USA. [2] Department of Neurological Surgery, University of Washington, 1900 9th Avenue, JMB10, Seattle, WA 98101, USA. [3] Department of Pediatrics, University of Washington, 1900 9th Avenue, JMB10, Seattle, WA 98101, USA. Correspondence and requests for materials should be addressed to J.M.R. (email: jan.ramirez@seattlechildrens.org)

Neural rhythmicity plays a key role in orchestrating critical brain functions ranging from consciousness, attention, perception, and memory[1–5] to feeding, locomotion, and breathing[6–9]. Rhythmicity often emerges from neural networks containing a combination of excitatory and inhibitory neurons. However, it has been challenging to unravel how the dynamic balance between excitation and inhibition influences the characteristic frequency of an ongoing rhythm.

An ideal model system for studying rhythmicity is the pre-Bötzinger Complex (preBötC), an autonomous oscillator located bilaterally in the ventrolateral medulla that generates the inspiratory phase of breathing[10]. The preBötC is highly amenable for rigorous electrophysiological investigation in vitro and has a clear behavioral correlate for in vivo studies. An essential feature of breathing is that it is dynamic and can operate over a broad frequency range. For example, in mice, breathing can occur at frequencies near 0.1 Hz during gasping[11] but can increase by two orders of magnitude to 11 Hz during sniffing[12,13]. How the preBötC generates rhythmicity across such a large and dynamic frequency range poses an important, yet unresolved question.

The preBötC is a heterogeneous network that contains multiple subpopulations of molecularly defined neurons and receives a rich array of neuromodulatory and sensory inputs[14–16]. Glutamatergic interactions allow preBötC neurons to synchronize and are therefore obligatory for rhythmogenesis. However, an estimated half of all preBötC neurons are inhibitory[17–19], and their role in rhythmogenesis is a topic of debate[20–23].

Among preBötC glutamatergic neurons, a subset is derived from precursors that express the transcription factor developing brain homeobox 1 protein (Dbx1) during development (referred to here as "Dbx1 neurons")[24–26]. These neurons are thought to have enhanced burst-generating properties and excitability[27] and therefore be the drivers (i.e., have pre-inspiratory activity) of the preBötC rhythm[27,28]. However, in rhythmically active brainstem slices containing the preBötC, optogenetic stimulation of Dbx1 neurons closely following a population burst cannot evoke a subsequent burst[29]. This refractory period suggests that Dbx1 neurons have a limited ability to drive high frequency bursting. How the preBötC network overcomes the limitations of the refractory period to produce rapid and dynamic breathing rhythms characteristic of behaving animals is unknown.

Here we combine targeted optogenetic manipulations with electrophysiology in vitro and in vivo to explore how subpopulations of excitatory and inhibitory preBötC neurons interact to control breathing frequency. We demonstrate that hyperactivity of excitatory neurons results in an intrinsic refractory period for preBötC bursting, while inhibitory neurons integrated within the active network limit hyperactivity and the refractory period, thereby permitting rapid and flexible breathing frequencies. We propose a preBötC architecture in which inhibition opposes rhythmogenesis, yet is critical for normal breathing in the intact animal. These results may extend beyond breathing and have important implications for understanding mechanisms of rhythm generation in general.

## Results

**Expression of Cre-dependent ChR2 in the preBötC.** To manipulate specific molecularly defined preBötC neuronal populations (Fig. 1a), mice expressing Cre recombinase under the *Dbx1*, vesicular glutamate transporter 2 (*Vglut2*), or vesicular GABA/glycine transporter (*Vgat*) promoter were crossed with mice containing a loxP-flanked STOP cassette prior to a sequence for a ChR2-reporter fusion protein. Thus ChR2-reporter was selectively expressed in cells also expressing Cre recombinase. A subset of Cre-driver mice were crossed with Cre-dependent

ZsGreen reporter mice to better represent the distribution of Cre + cell bodies rather than the membrane bound ChR2-reporter (Fig. 1b–e). Since "Dbx1 neurons" within the preBötC express the *Dbx1* gene transiently during embryonic development, a tamoxifen-inducible Cre (*Dbx1CreERT2*) was used and pregnant dams were given a tamoxifen pulse at E10.5 to induce Cre expression[24,26]. Similar to previous reports utilizing *Dbx1CreERT2*[25,30,31], reporter labeling was observed in transverse medullary slices within a well-defined region extending ventrolaterally from the hypoglossal (XII) motor nucleus encompassing the intermediate reticular formation (IRt), a region containing XII premotor neurons[32], and the preBötC just ventromedial to ChAT+ nucleus ambiguus neurons (Fig. 1b). Although *Dbx1CreERT2* preferentially labels neurons and glia in the preBötC and IRt and has been used in the majority of studies defining the role of Dbx1 neurons in respiratory rhythmogenesis (e.g., refs. [26,27,29,33–35]), *Dbx1CreERT2* will not label all Dbx1-derived preBötC neurons[30]. Therefore, we combined this approach with use of the *Vglut2Cre* line that specifically labels all glutamatergic preBötC neurons, including the vast majority Dbx1 neurons[24,26]. As expected, reporter fluorescence in *Vglut2Cre* and *VgatCre* mice was widely distributed within the medulla. However, strong reporter expression was observed in cell bodies within the preBötC of all animals from each experimental group (Fig. 1c–e), reflecting the heterogeneity of neuron subtypes within the preBötC.

**An intrinsic refractory period for preBötC neuron activity.** To explore mechanisms regulating preBötC bursting, we recorded preBötC population activity in slices from *Dbx1*-ChR2 and *Vglut2*-ChR2 neonatal (P4–12) mice (Fig. 2a and Supplementary Fig. 1a). Excitatory neurons were stimulated contralateral to population recordings with 200 ms blue light pulses and the probability of light evoking a population burst was plotted against elapsed time following an endogenous preBötC burst (Fig. 2b). Confirming previous results[29], we observed a refractory period of ~2 s following endogenous preBötC bursts during which stimulation of Dbx1 or Vglut2 neurons did not evoke a burst (Fig. 2a, b). Intracellular current-clamp recordings of Dbx1 neuron activity within the active preBötC network revealed that the magnitude of depolarization during inspiratory bursts was proportional to a membrane afterhyperpolarization (AHP) that followed[36] (Fig. 2c), suggesting that intrinsic properties of preBötC neurons may play a significant role in the refractory period mechanism. To test this, rhythmically active preBötC neurons were synaptically isolated pharmacologically with antagonists of AMPA, *N*-methyl-D-aspartate, glycine, and GABA receptors (CNQX, CPP, strychnine, and gabazine, respectively) and current was injected to simulate a burst "drive potential". Reminiscent of Dbx1 neuron activity within the network, isolated Dbx1 neurons were hyperpolarized following current injection, and the AHP depended on the intensity of the depolarizing current step (Fig. 2d). To examine how intrinsic AHP translates to a network effect, we quantified spiking activity during small (20 pA) consecutive current pulses following a larger (20–60 pA) depolarizing current step (Fig. 2e). The number of spikes evoked with each small current pulse was reduced 0.75–1.25 s following the initial larger depolarizing pulse and returned to near maximal levels by 2.25–2.75 s, corresponding to the duration of the refractory period for preBötC population activity (see Fig. 2b). The reduction in spiking activity was related to the intensity (Fig. 2e) and duration (Supplementary Fig. 2a) of the simulated drive potential. Synaptically isolated neurons were identified as Dbx1+ with depolarizing responses to light. Notably, spiking was also reduced following current injection in isolated Dbx1−

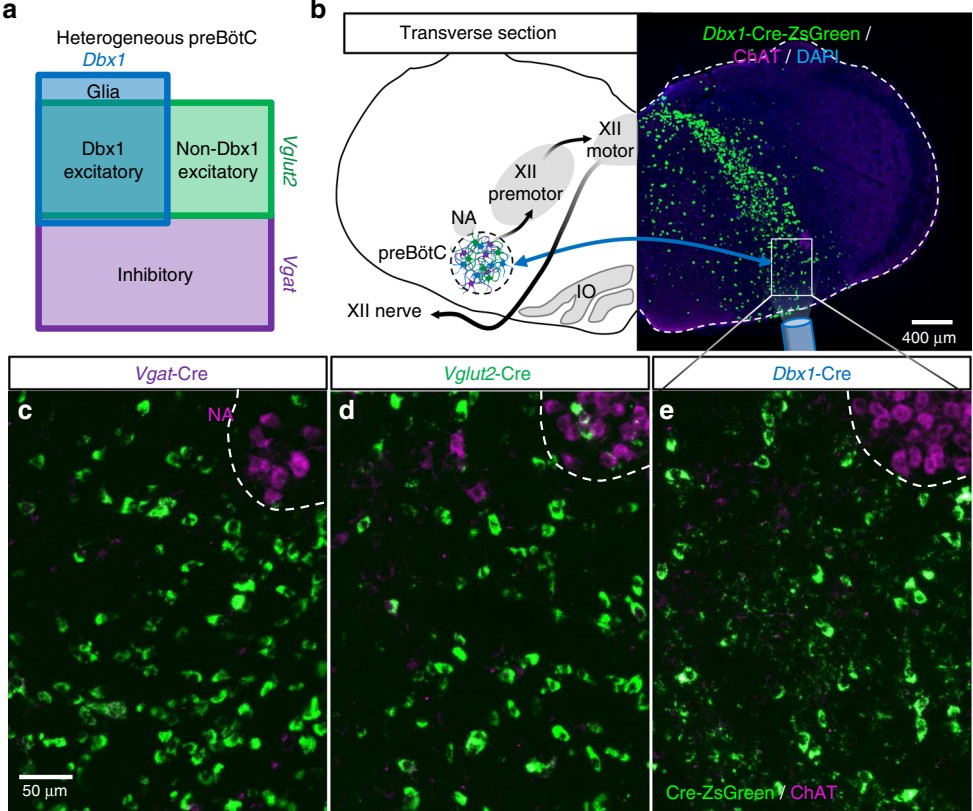

**Fig. 1** Anatomy of the heterogeneous preBötC. **a** Diagram showing estimated ratios of Dbx1 and non-Dbx1 excitatory and inhibitory neurons[19, 24–26] with the corresponding promoters used to drive Cre expression within these preBötC subpopulations. **b** Schematic depicting key landmarks (NA nucleus ambiguus, IO inferior olive) and the pathway from preBötC rhythmogenesis to XII motor output and a representative 2.5× image of a transverse medullary preBötC section from an adult *Dbx1-ZsGreen* mouse. Blue arrows represent commissural connections between Dbx1 neurons[24]. **c–e** 20× z-projected images of Cre-dependent ZsGreen expression in preBötC neurons from *Vgat^Cre* (**c**), *Vglut2^Cre* (**d**), and *Dbx1^CreERT2* (**e**) mice. ChAT immunofluorescence demarks the NA dorsomedial to the preBötC

neurons (Supplementary Fig. 2b), suggesting refractory mechanisms may be similar for Dbx1+ and Dbx1− preBötC neurons.

**Inhibition modulates the refractory period**. Next, we examined how inhibitory preBötC neurons influence the activity of excitatory Dbx1 preBötC neurons and rhythmogenesis. Since the preBötC contains both glycinergic[18,19] and GABAergic[17] neurons, we recorded from rhythmic preBötC neurons in *Dbx1-ChR2* slices during progressive bath application of strychnine (1 μM) and gabazine (1 μM) to block glycinergic and GABAergic fast synaptic inhibition, respectively (Fig. 3a). Strychnine and gabazine increased drive potential area (Fig. 3a) as well as the amplitude of preBötC population bursts (Fig. 3b). Thus both glycinergic and GABAergic mechanisms restrain Dbx1 neuron activity during synchronized population activity.

To determine how inhibition influences the refractory period for preBötC population bursting, we delivered light pulses randomly during the respiratory cycle while recording contralateral preBötC population activity under baseline conditions and after blocking glycinergic and then GABAergic, synaptic transmission (Fig. 3b). The probability of light-evoking a population burst was plotted against elapsed time following an endogenous preBötC burst (Fig. 3c). Under baseline conditions, a burst was evoked ~50% of the time 1.08 s and ~100% of the time 2.75 s following an endogenous burst. When glycinergic inhibition was blocked, a burst was evoked ~50% of the time at 2.25 s and ~100% of the time at 3.75 s. After blocking glycinergic and

GABAergic inhibition, there was a ~50% chance of evoking a burst 3.80 s and a ~100% chance 5.75 s following an endogenous preBötC burst. The change in cycle duration (phase shift) relative to the time of light onset was also assessed (Fig. 3d, Supplementary Fig. 1b), revealing similar results. Thus both glycinergic and GABAergic inhibition reduce the refractory period for preBötC bursting. Further, we found that longer refractory periods were associated with slower preBötC rhythms since preBötC burst frequency was reduced during blockade of inhibition (Fig. 3e). Note, that in all conditions (baseline, strychnine, and strychnine +gabazine), the end of the refractory period corresponded with an increased probability of an endogenous burst occurring.

Similar mechanisms also apply to sighs, large amplitude biphasic bursts generated by reconfiguration of the preBötC network[37,38] that are followed by a "post-sigh apnea"[39]. We found that Dbx1 neuron drive potential, AHP, and the refractory period were increased during sigh bursts and relative to normal "eupneic" bursts (Supplementary Fig. 3). Thus we posit that the post sigh apnea results, at least in part, from an exaggerated refractory period due to increased preBötC activity during sighs.

**Phase-dependent activity of preBötC inhibitory neurons**. Next, we explored the activity of inhibitory preBötC neurons in *Vgat*-ChR2 slices (Fig. 4a). Individual neurons were identified as Vgat+ with a depolarizing response to blue light, whereas Vgat− neurons typically hyperpolarized, followed by postinhibitory rebound[40] coincident with a preBötC population burst at the end

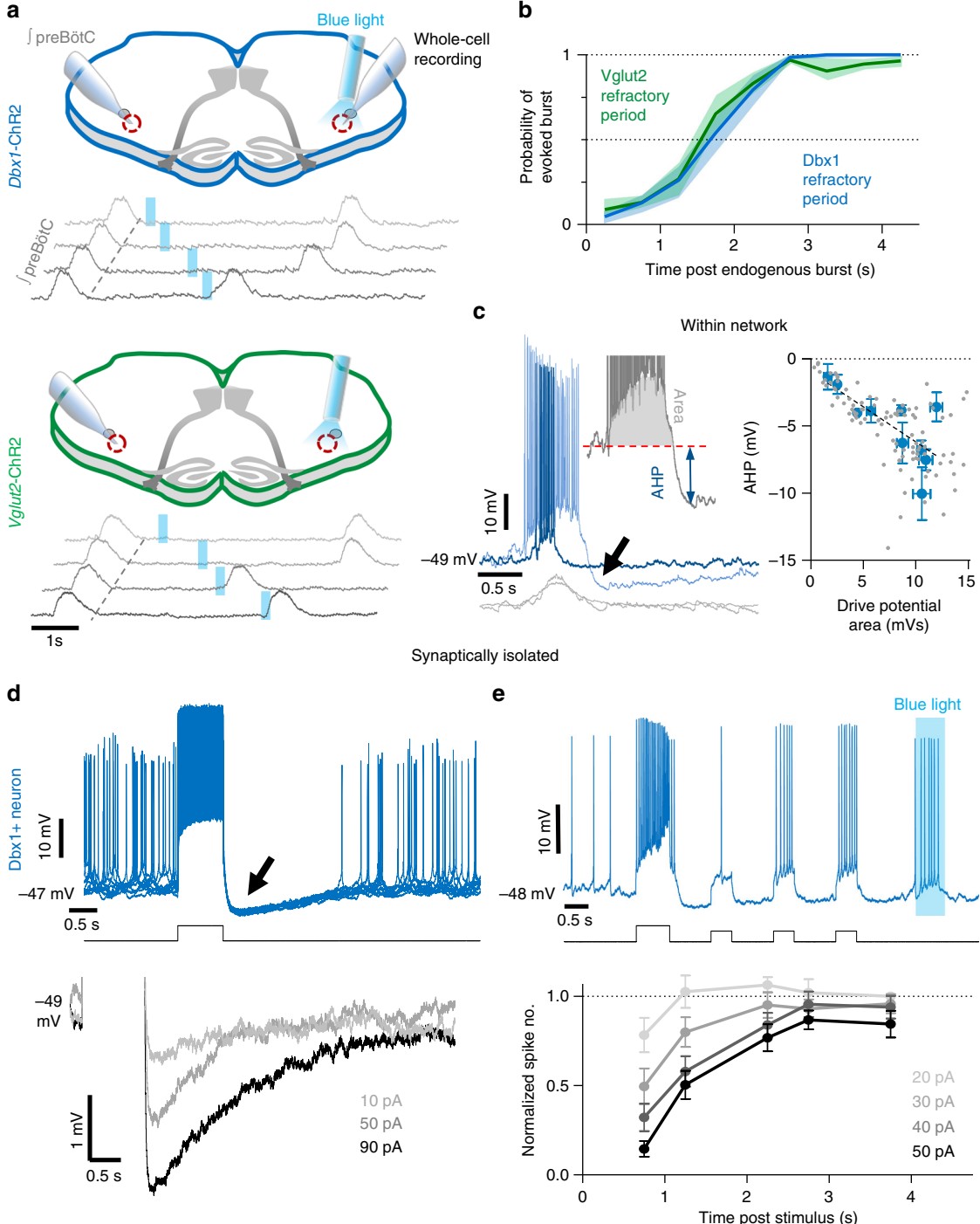

**Fig. 2** Intrinsic refractory properties of preBötC neurons. **a** Diagram of in vitro brainstem slice preparation and example of the refractory period for evoked preBötC population bursts in a *Dbx1*-ChR2 and *Vglut2*-ChR2 slice. **b** Quantified probability of light-evoking a preBötC burst by stimulation excitatory neurons in *Dbx1*-ChR2 ($n = 7$) and *Vglut2*-ChR2 ($n = 7$) slices. ($p > 0.05$ at all time points, two-way ANOVA and Bonferroni's multiple comparisons test, means ± s.e.m. **c** Overlaid activity of two Dbx1+ neurons in the spontaneously active preBötC network demonstrating the relationship between burst drive potential and AHP (arrow). Quantified data from $n = 10$ Dbx1+ neurons shown on the right (linear regression analysis, $p < 0.0001$; slope $= -0.52 \pm 0.058$). **d** Representative example (10 overlaid sweeps) of a synaptically isolated Dbx1+ neuron showing hyperpolarization (arrow) and reduced spiking following a 600 ms 50 pA current step (upper). Membrane potential (average of 10 sweeps) before and after graded current step intensities (bottom). **e** Example showing transiently reduced number of spikes evoked by small (20 pA, 400 ms) current steps during the intrinsic refractory period following a 30 pA current step in a synaptically isolated Dbx1+ neuron (note the depolarization and spiking evoked by light); average normalized number of spikes evoked at different time points following current steps of varied intensity ($n = 7$ neurons from $n = 7$ slices; means ± s.e.m.)

of light stimulation (Fig. 4b). Inhibitory preBötC neurons rhythmically active during the inspiratory phase were abundant[18]. Fourteen out of the 37 (37%) recorded inspiratory neurons were Vgat+ (Fig. 4b), although none had pre-inspiratory activity (Supplementary Fig. 4). PreBötC neurons active during the expiratory phase were much less common[41]. However, all recorded expiratory neurons (10/10) were Vgat+ (Fig. 4c). Expiratory neurons became inspiratory following blockade of synaptic inhibition (Fig. 4c), indicating that they receive concurrent excitation and inhibition, with inhibition dominating under control conditions[42–44]. Indeed, some inhibitory neurons were silenced during small preBötC bursts but excited during larger bursts or sighs[45] (Supplementary Fig. 5).

**Phasic inhibition elicits rapid preBötC bursting**. To examine phase-dependent effects of inhibitory neurons on the ongoing

preBötC rhythm, we unilaterally or bilaterally stimulated Vgat+ neurons specifically during inspiration or expiration with light pulses threshold-triggered by preBötC population bursts. While blocking synaptic inhibition slowed the rhythm (see Fig. 3b, e), increasing inhibition optogenetically during inspiration increased preBötC burst frequency (Figs. 4d and 5b, c). Bilateral light stimulation elicited stable and robust high frequency bursting (314 ± 46% increase to 1.37 ± 0.16 Hz), whereas bursting was more irregular and slower on average during unilateral stimulation (93 ± 16% increase to 0.90 ± 0.08 Hz). Importantly, frequency was increased beyond the maximum possible frequency predicted by the refractory period (1 burst/~2 s or 0.5 Hz). During bilateral stimulation of Vgat+ neurons, increases in burst frequency were laser power-dependent, suggesting frequency can be tuned by the amount of inhibitory neuron activation (Supplementary Fig. 6a). These effects were blocked in strychnine and gabazine (Supplementary Fig. 6b). Unlike phasic stimulation during inspiration, activation of inhibitory neurons during the expiratory phase slowed preBötC bursting by −37 ± 7% during bilateral stimulation and −26 ± 4% during unilateral stimulation (Figs. 4d and 5c). Thus inhibition has phase-dependent effects on respiratory frequency[23], and reducing the refractory period with inhibition allows a rapid preBötC rhythm.

Next, we compared these results with the effects of phasic stimulation of excitatory neurons. Dbx1 neuron stimulation during inspiration had very little effect on frequency, with decreases of −10 ± 3% elicited during bilateral stimulation and −10 ± 3% during unilateral stimulation (Fig. 5c). Targeting light stimulation specifically during the expiratory phase only moderately increased burst frequency (Fig. 5a, c). Bilateral stimulation increased frequency by 32 ± 15% to 0.41 ± 0.05 Hz and unilateral stimulation by 23 ± 11% to 0.37 ± 0.03 Hz, levels similar to that predicted by the refractory period. Changes in burst frequency were similar for bilateral and unilateral stimulation of excitatory neurons, and phase-specific stimulations in *Vglut2*-ChR2 slices yielded similar results (Supplementary Fig. 6c).

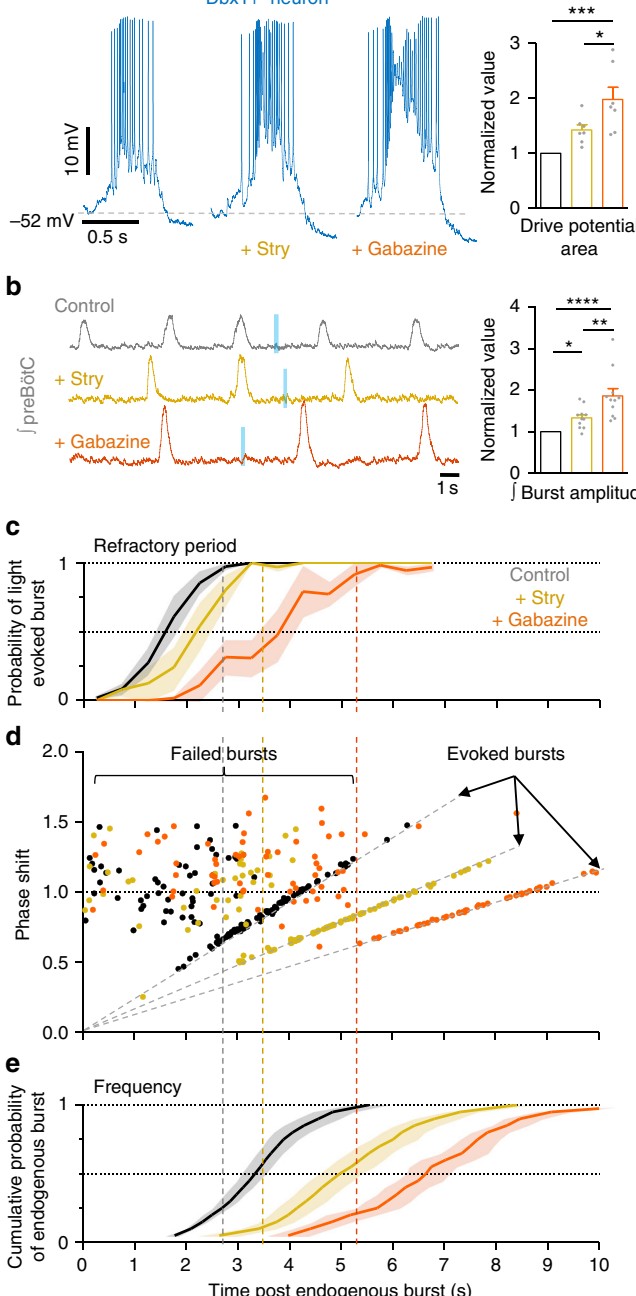

**Fig. 3** Inhibition restrains excitatory preBötC neuron activity and the refractory period. **a** Example Dbx1+ neuron activity under baseline conditions and following progressive blockade of glycinergic and gabaergic synaptic inhibition with strychnine (1 µM) and gabazine (1 µM), respectively. Average normalized inspiratory drive potential area (n = 7) of Dbx1+ neurons during preBötC bursts in strychnine and gabazine (*p < 0.05, ***p<0.001; means ± s.e.m.; one-way repeated measures ANOVA and Bonferroni's multiple comparisons test). **b** Example of preBötC population activity from a *Dbx1*-ChR2 slice during inhibition block and failure to light-evoke bursts during the refractory period, and normalized burst amplitude of integrated preBötC population activity (n = 11 slices; *p < 0.05, **p<0.01, ****p<0.0001; means ± s.e.m.; Friedman test and Dunn's multiple comparisons test). **c** Quantified probability of evoking a burst (0.5 s bins) relative to time post endogenous burst (s) from *Dbx1*-ChR2 slices during inhibition block (~100–150 trials in each condition from each slice; n = 4 slices; means ± s.e.m.). **d** Data from a representative *Dbx1*-ChR2 stimulation experiment comparing the phase shift (stimulus cycle duration/average cycle duration) elicited compared to the time of the light stimulus relative to the preceding endogenous burst. Note that in strychnine (yellow) and strychnine+gabazine (orange) failed bursts are more common for a longer duration following endogenous bursts (358 stimulations). Average data shown in Supplementary Fig 1. **e** Cumulative probability of a spontaneous burst (0.05 probability bins) relative to time post endogenous burst (s) from *Dbx1*-ChR2 slices during inhibition block (~100–150 trials in each condition from each slice; n = 4 slices; means ± s.e.m.). Vertical dashed lines correspond to the end of the refractory period and an increasing probability of spontaneous bursting

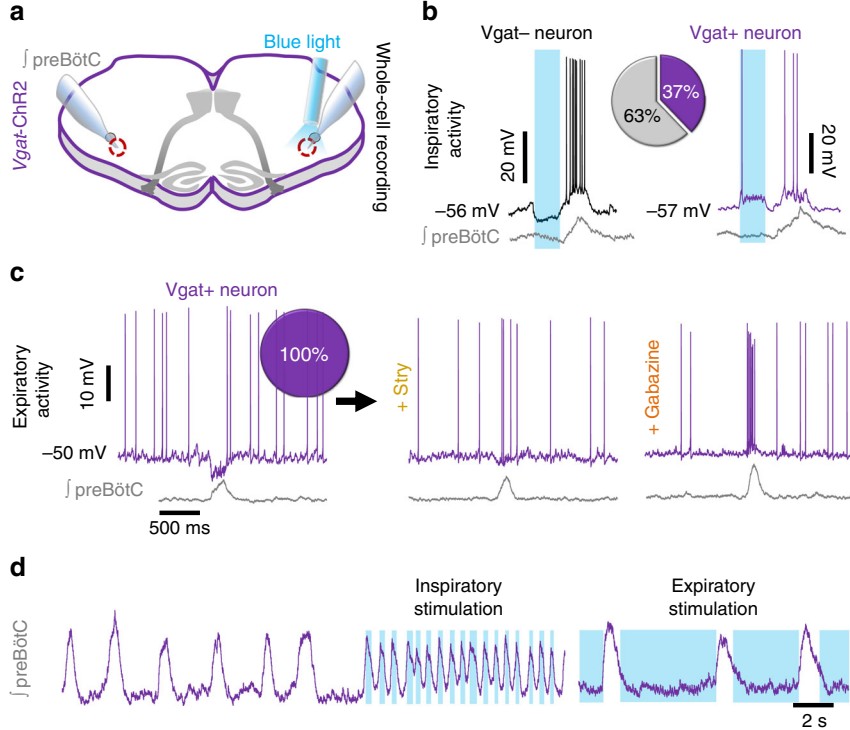

**Fig. 4** PreBötC inhibitory neuron populations have distinct effects on inspiratory burst frequency. **a** Diagram of *Vgat*-ChR2 brainstem slice configuration used to identify inhibitory preBötC neuron activity relative to preBötC burst phase. **b** Representative activity of a Vgat− inspiratory neuron showing post-inhibitory rebound following light stimulation (left) and a Vgat+ neuron with inspiratory activity (right). 14/37 (37%) recorded inspiratory neurons were inhibitory. **c** Example Vgat+ expiratory neuron at baseline and during blockade of synaptic inhibition with strychnine (1 µM) and gabazine (1 µM) (*n* = 3). Note that expiratory activity transitions to inspiratory activity following blockade of synaptic inhibition. 10/10 (100%) recorded expiratory neurons were inhibitory. **d** Example recording of preBötC population activity from a *Vgat*-ChR2 slice during phase-specific light stimulation. Average data shown in Fig. 5c

Since postinhibitory rebound can trigger preBötC bursts[40] (see Fig. 4b), we sought to determine whether this mechanism could overcome the refractory period to drive rapid preBötC bursting. Using *Vgat*-ChR2 slices, we optically stimulated preBötC inhibitory neurons with light pulses randomly during the respiratory cycle and calculated the probability of evoking a population burst via postinhibitory rebound (Fig. 5d) and the phase shift (Supplementary Fig. 1a). Rebound bursts were rarely generated during the first ~2 s following an endogenous preBötC burst, indicating a refractory period for postinhibitory rebound very similar to that determined for Dbx1 stimulation (Fig. 5d). Thus unless the refractory period is reduced by limiting the activity of excitatory neurons during inspiration, rebound mechanisms per se cannot overcome the refractory period to drive rapid bursting.

**PreBötC inhibition is partially mediated by vagal afferents**. To explore how excitatory and inhibitory mechanisms contribute to the control of breathing frequency in the intact respiratory network where frequencies are typically much faster (Supplementary Fig. 7), we recorded inspiratory activity from the hypoglossal nerve in anesthetized, spontaneously breathing, optogenetic adult mice with open access to the ventral medullary surface (Fig. 6a). Optogenetic stimulations of preBötC subpopulations were performed before and after transection of the vagal nerves, resulting in a constant slow breathing rate[46]. Vagal afferents respond to lung stretch and have a phasic inhibitory influence on breathing to prevent over inflation of the lungs (i.e., Breuer–Hering reflex)[47]. To confirm that transection of the vagal nerves reduces

inhibition locally within the preBötC, we compared the effects of vagotomy with bilateral nanoinjection of strychnine (250 µM) and gabazine (250 µM) into the preBötC (Fig. 6b–e). Similar to Janczewski et al.[20], but see Marchenko et al.[21], both vagotomy and inhibition blockade slowed, but did not stop, breathing and increased XII amplitude. Changes in frequency and XII amplitude were negatively correlated (linear regression, *p* < 0.0001). Slowed frequency was primarily due to an increase in expiratory time (Te) (154 ± 20%); however, inspiratory time (Ti) was also increased (55 ± 5%). Importantly, nanoinjection of strychnine and gabazine into the preBötC had a larger effect on XII frequency and amplitude than vagotomy (Fig. 6e). Thus, consistent with our results in vitro (see Figs. 3 and 4b), not all preBötC inhibition is mediated by sensory feedback. Application of continuous positive airway pressure to induce lung stretch confirmed that the Breuer–Hering reflex was eliminated following blockade of preBötC inhibition[20] (Supplementary Fig. 8). Therefore, activity of vagal afferents ultimately inhibits breathing via inhibitory mechanisms within the preBötC.

Injection and photostimulation sites were marked with flourospheres (Fig. 6c, f). We found the optimal location to target the preBötC was 1.35 mm (48%) lateral from the basilar artery, 1.29 mm (47%) caudal to the caudal cerebellar artery, and 0.30 mm (12%) rostral to the intersection of the vertebral arteries. Extracellular recordings in this area revealed inspiratory population activity that occurred prior to XII bursts (Fig. 6f). Imaging of flourospheres injected at the stimulation site at the end of each experiment confirmed that there were no differences in the stimulation sites between experimental groups (Fig. 6g).

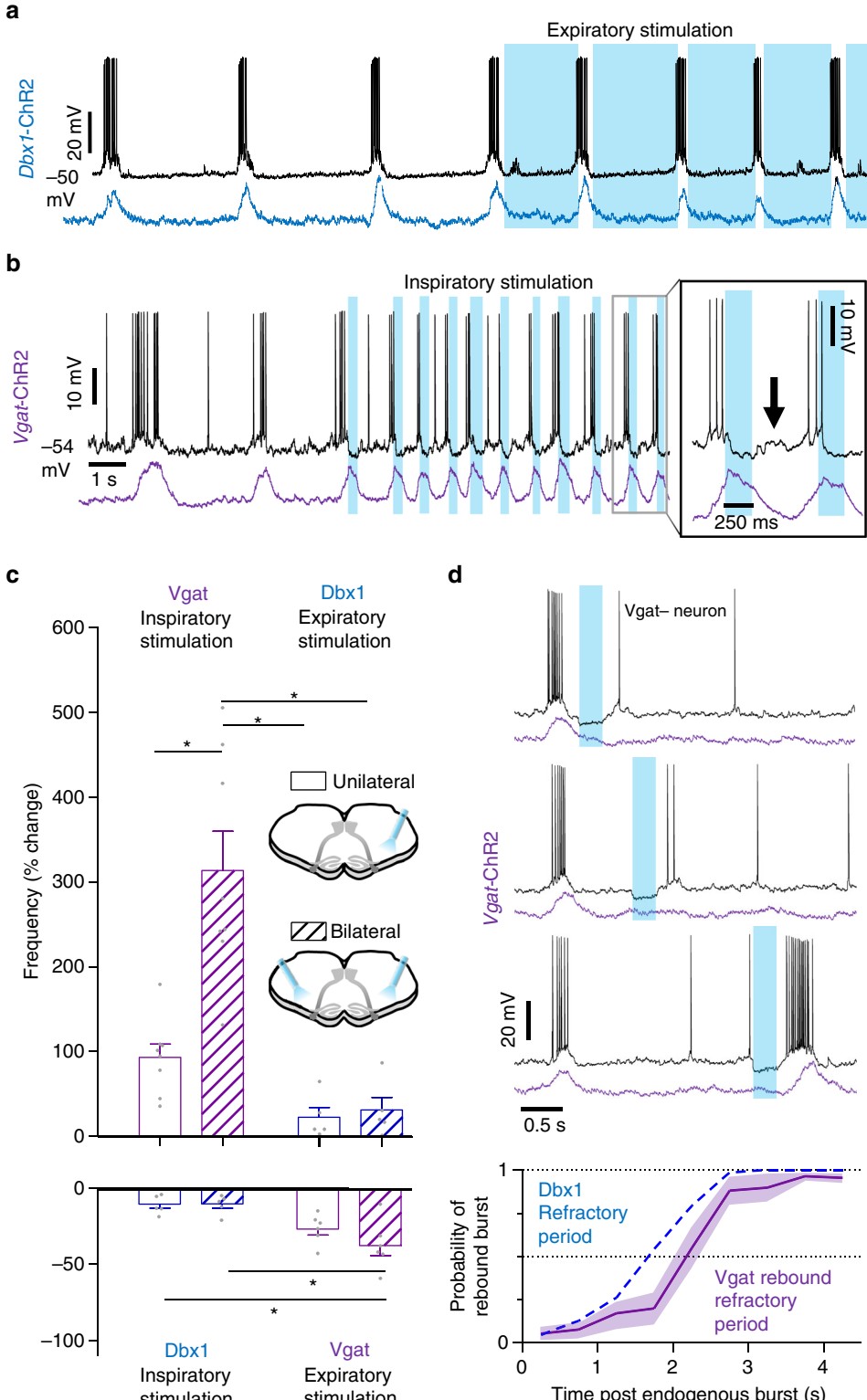

**Fig. 5** Phasic inhibition, but not excitation, drives high preBötC burst frequencies. **a** Representative inspiratory neuron and preBötC population recording during bilateral light stimulation specifically during the expiratory phase in a *Dbx1*-ChR2 rhythmic brainstem slice. **b** Representative inspiratory neuron and preBötC population activity during threshold-triggered bilateral light stimulation specifically during the inspiratory phase in a *Vgat*-ChR2 slice. **c** Average changes in frequency in *Vgat*-ChR2 slices during unilateral and bilateral preBötC stimulation during inspiration ($n = 8$) and expiration ($n = 6$) compared to *Dbx1*-ChR2 stimulation during inspiration ($n = 5$) and expiration ($n = 5$; one-way ANOVA and Bonferonni's multiple comparisons test; *$p < 0.001$). **d** Intracellular recording of a Vgat− neuron and preBötC population activity during brief light pulses (200 ms) in a *Vgat*-ChR2 slice. Note that rebound spiking is reduced during the refractory period. Quantified probability of evoking a population burst via postinhibitory rebound relative to the time of light stimulation following an endogenous burst (0.5 s bins; $n = 7$ slices, ~100–150 trials/slice; mean ± s.e.m.) compared to the probability of evoked bursts during Dbx1 stimulation (data shown in Fig. 2b), demonstrating similar refractory periods

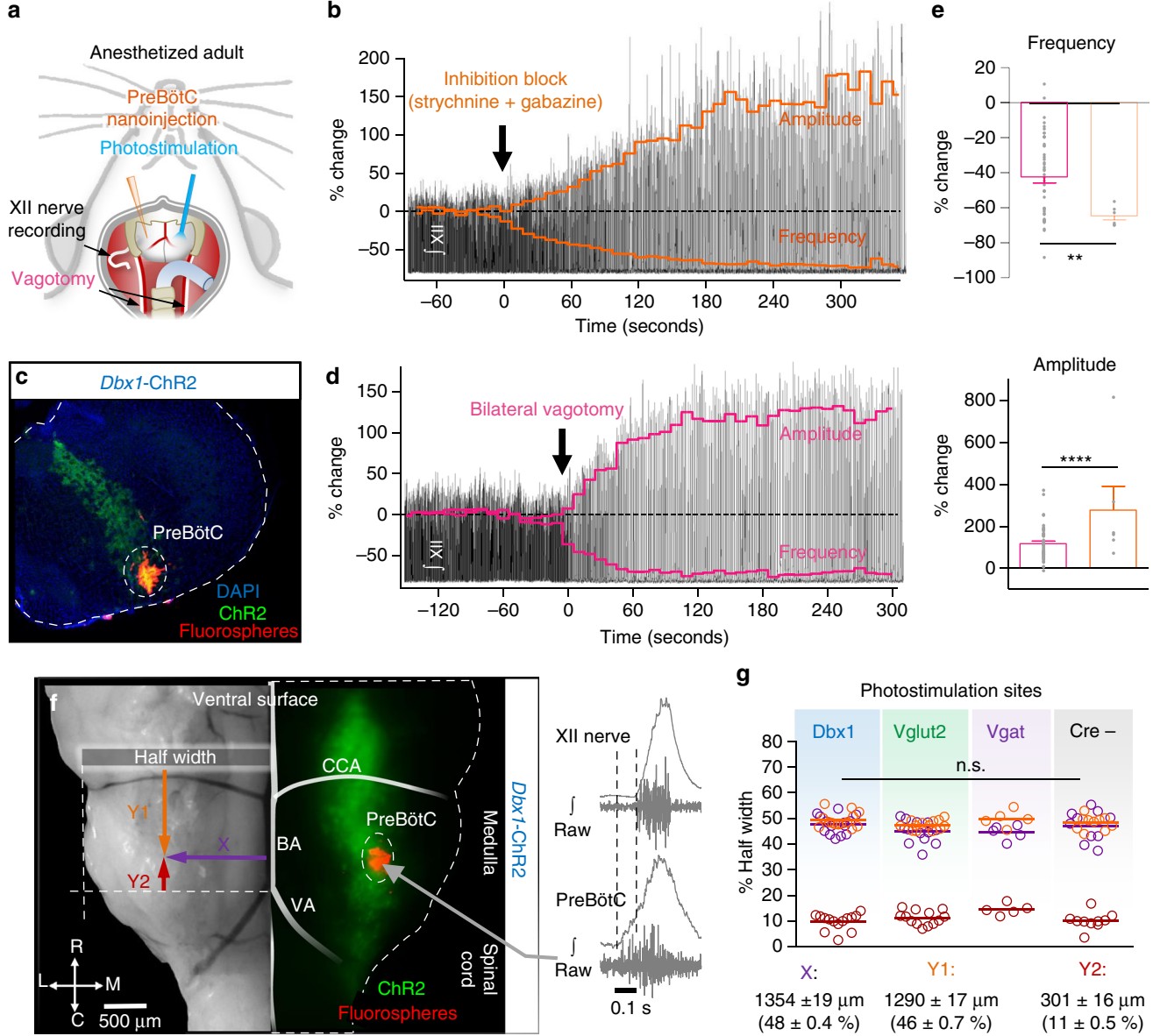

**Fig. 6** Effects of preBötC sensory feedback inhibition on hypoglossal motor output in vivo. **a** Schematic of the surgical approach to access the ventral brainstem for bilateral preBötC nanoinjections, photostimulation, and vagotomy. **b** Representative trace of integrated hypoglossal (XII) nerve activity during bilateral injection of strychnine and gabazine (150 nl, 250 μM each) with frequency and amplitude overlaid (10 s bins). **c** Example transverse hemisection showing injection site marked by flourospheres localized to the preBötC in a *Dbx1*-ChR2 mouse. **d** Representative trace of XII activity during bilateral vagotomy with frequency and amplitude overlaid (10 s bins). **e** Quantified changes in frequency and amplitude ~5 min following vagotomy ($n = 47$) and blockade of preBötC fast synaptic inhibition ($n = 6$; unpaired two-tailed *t*-tests with Welch's correction; **$p < 0.01$, ****$p < 0.0001$). **f** Bright field image (left) and *Dbx1*-ChR2 fluorescence (middle) of the ventral medulla showing the location of preBötC photostimulation relative to the basilar artery (BA), caudal cerebellar artery (CCA), and intersection of the vertebral arteries (VA). Example extracellular recording demonstrating pre-inspiratory activity relative to XII nerve activity at photostimulation sites (right). **g** Quantified coordinates of injected fluorospheres (bottom) demonstrate that photostimulation sites were consistent across experimental groups (Dbx1, $n = 14$; Vglut2, $n = 14$; Vgat, $n = 5$; Cre-, $n = 10$; two-way ANOVA and Bonferonni's post hoc test; n.s., not significant, $p > 0.05$)

**Limitations on excitatory preBötC mechanisms in vivo**. To test the ability of excitatory mechanisms to generate rapid breathing in vivo, we recorded hypoglossal activity during a continuous 10 s photostimulation of overlapping glutamatergic *Dbx1*-ChR2 and *Vglut2*-ChR2 subpopulations (Fig. 7). Light stimulation in vagus-intact *Dbx1*-ChR2 mice increased XII nerve burst amplitude (31 ± 3%) but, surprisingly, had almost no effect on breathing frequency (5 ± 2%). In contrast, stimulation in *Vglut2*-ChR2 mice elicited a moderate increase in both breathing frequency (23 ±

3%) and XII amplitude (24 ± 3%) (Fig. 7a, c). During slow breathing following loss of vagal sensory feedback, continuous light stimulation of the preBötC moderately increased breathing frequency in both *Dbx1*-ChR2 (24 ± 5%) and *Vglut2*-ChR2 (21 ± 5%) mice (Fig. 7b, d). Effects on XII amplitude during preBötC photostimulation were abolished in vagotomized mice, indicating that vagotomy minimizes dynamic amplitude responses[28].

Under both vagus-intact and vagotomized conditions, changes in breathing frequency were reflected in reduced Te (Fig. 7e,f).

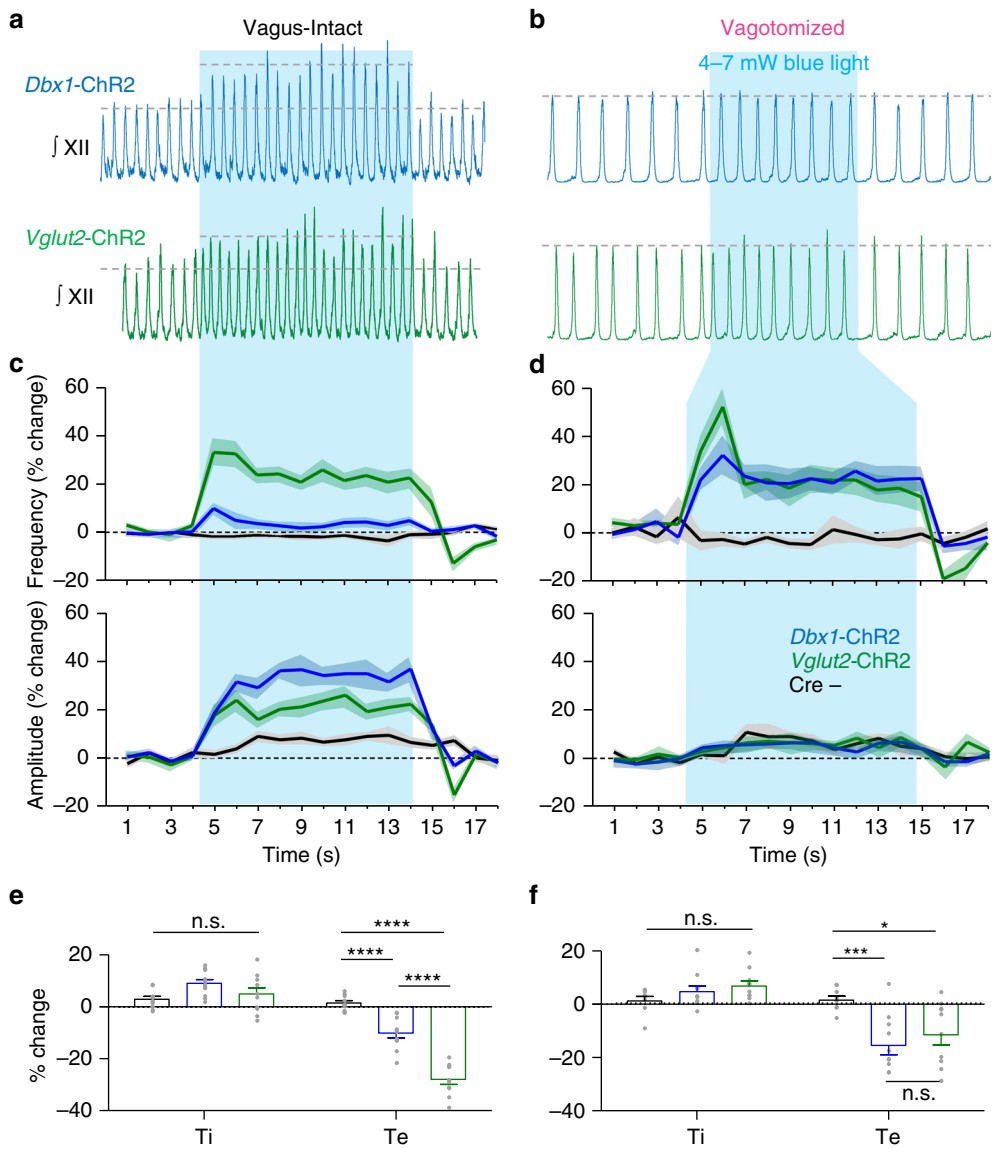

**Fig. 7** Excitatory preBötC neurons have a limited ability to generate rapid breathing in vivo. **a, b** Representative hypoglossal (XII) nerve activity during a continuous 10 s stimulation of the preBötC in *Dbx1*-ChR2, *Vglut2*-ChR2, and Cre- control mice with the vagus intact (**a**) and following vagotomy (**b**). **c, d** Average XII nerve burst frequency and amplitude (1 s bins) during continuous light stimulation with the vagus intact (**c**) (Dbx1, $n = 11$; Vglut2, $n = 10$; Cre−, $n = 10$) and following vagotomy (**d**) (Dbx1, $n = 10$; Vglut2, $n = 10$; Cre−, $n = 8$; means ± s.e.m.). **e, f** Average changes in inspiratory and expiratory time (Ti and Te, respectively) during continuous stimulation in vagus-intact (**e**) and vagotomized (**f**) mice demonstrating that changes in breathing frequency are primarily the result of reduced Te (mean ± s.e.m.; two-way ANOVA and Bonferonni's multiple comparisons test; \*\*$p < 0.01$, \*\*\*$p < 0.001$, \*\*\*\*$p < 0.0001$)

Results were similar across increasing laser powers (Supplementary Fig. 9). Consistent with our results in vitro (see Fig. 5c), unilateral and bilateral stimulations of excitatory neurons had similar, moderate effects on breathing frequency (Supplementary Fig. 9a), suggesting that there is a ceiling effect for increased breathing frequency elicited by activation of excitatory preBötC neural populations.

**Vagal feedback differentially modulates excitatory neurons**. To further investigate the differential frequency response observed during continuous stimulation of Dbx1 and Vglut2 neurons, we selectively stimulated the preBötC during the inspiratory or expiratory phase with light pulses threshold-triggered by XII nerve bursts (Fig. 8). With the vagus intact, selective stimulation during inspiration in both *Dbx1*-ChR2 and *Vglut2*-ChR2 mice elicited a similar increase in XII amplitude (Dbx1: 22 ± 5%,

Vglut2: 19 ± 3) and decrease in breathing frequency (Dbx1: −13 ± 1%, Vglut2: −11 ± 1%) primarily via prolonged Te (Dbx1: 24 ± 1%, Vglut2:16 ± 4%) (Fig. 8a, c). Thus, with vagal feedback intact, increasing the activity of excitatory preBötC neurons during an inspiratory burst delays the subsequent burst and slows breathing. In contrast, during expiratory stimulation, breathing frequency was increased (Dbx1: 15 ± 2%, Vglut2: 28 ± 4%) via a reduction in Te (Dbx1: −24 ± 2%, Vglut2: −42 ± 3%), and the effect was greater in *Vglut2*-ChR2, compared to *Dbx1*-ChR2 mice (Fig. 8a, c). It is notable that, for Dbx1 stimulation, the amount frequency was reduced during inspiratory stimulation was nearly equivalent to the amount it was increased during expiratory stimulation, likely leading to the minimal frequency effect of continuous stimulation.

During slow breathing following vagotomy[46], phase-specific stimulation of either excitatory population during inspiration no longer had an effect on XII frequency or amplitude, suggesting

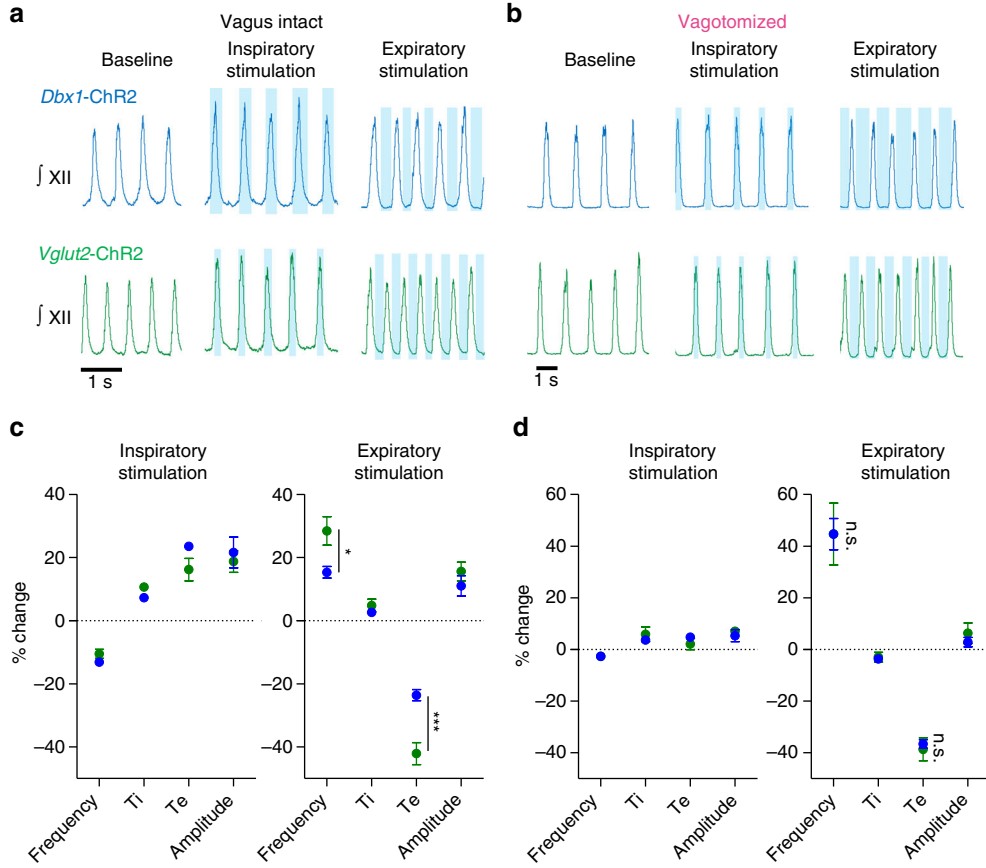

**Fig. 8** Differential frequency control of overlapping excitatory preBötC populations is specific to expiratory stimulation in vagus-intact, but not in vagotomized, mice. **a, b** Representative XII nerve activity during respiratory phase-specific inspiratory or expiratory preBötC stimulation in *Dbx1*-ChR2 and *Vglut2*-ChR2 mice with the vagus intact (**a**) and following vagotomy (**b**). **c, d** Average changes in frequency, Ti, Te, and XII burst amplitude during inspiratory and expiratory stimulation in vagus-intact (**c**) (Dbx1, $n = 5$; Vglut2, $n = 6$) and vagotomized (**d**) (Dbx1, $n = 4$; Vglut2, $n = 4$) mice (mean ± s.e. m.; two-way ANOVA and Bonferonni's multiple comparisons test; $*p < 0.05$, $***p < 0.001$). Note that Vglut2 stimulation only elicits a larger frequency effect than Dbx1 stimulation when sensory feedback inhibition is intact (vagus intact) and when stimulation is specific to the expiratory phase

that excitatory neurons are nearly maximally active during preBötC bursts in vagotomized mice (Fig. 8b, d). Similar to results obtained in vagus-intact mice, breathing frequency was increased during expiratory stimulation (Dbx1: $45 \pm 6\%$, Vglut2: $45 \pm 11\%$) due to decreased Te (Dbx1: $-37 \pm 2\%$, Vglut2: $-39 \pm 4\%$). However, vagotomy eliminated the differential effect of *Dbx1*-ChR2 and *Vglut2*-ChR2 stimulation (Fig. 8b, d), suggesting vagal feedback differentially modulates the ability of preBötC excitatory populations to increase breathing frequency specifically during the expiratory phase. Despite increasing frequency by ~40% to $1.0 \pm 0.2$ Hz, stimulation of either excitatory preBötC subpopulation could not drive breathing to pre-vagotomy frequencies ($2.1 \pm 0.5$ Hz, $p < 0.0001$), further indicating that excitatory mechanisms alone have a limited ability to drive rapid breathing.

**Vagal feedback limits the refractory period**. We next sought to characterize the refractory period for breathing in vivo. Unlike in vitro stimulations where bursts were clearly evoked or not, stimulations in vivo typically resulted in a shift in the onset of the subsequent inspiratory burst. Therefore, the change in the cycle duration containing the stimulus relative to previous, unperturbed cycles (i.e., "phase shift") was used to assess the refractory period in vivo, which was defined as the time between the end of inspiration and the stimulus phase that elicited the maximum phase shift.

In vagus-intact *Dbx1*-ChR2 mice, we observed a refractory period of ~200 ms (Fig. 9a). However, in *Vglut2*-ChR2 mice, a

phase advance was elicited when stimulation occurred within the first 200 ms following the end of inspiration (Fig. 9a, c). Indeed, the phase shift elicited by *Vglut2*-ChR2 stimulation was significantly greater than the Dbx1 phase shift during the first 200 ms following inspiration ($p < 0.05$), while near the end of the respiratory cycle, *Vglut2*-ChR2 and *Dbx1*-ChR2 stimulation had equivalent effects ($p > 0.05$). Following vagotomy, the refractory period was increased and the differential effect of *Dbx1*-ChR2 and *Vglut2*-ChR2 stimulation was lost (Fig. 9b, d), such that stimulation did not result in a phase advance for a period of 500–600 ms following the end of inspiration in both *Dbx1*-ChR2 and *Vglut2*-ChR2 mice. Near the end of the refractory period, failed bursts were occasionally apparent (arrows in Fig. 9b); and if stimulations occurred later in the respiratory cycle, a phase advance was elicited. Thus loss of sensory feedback inhibition exaggerates the refractory period, thereby slowing breathing frequency.

**Inhibitory neurons promote rapid dynamic breathing**. Next, we examined the phase-dependent effects of stimulating inhibitory preBötC neurons in vivo by randomly delivering brief light pulses to the preBötC of *Vgat*-ChR2 mice (Fig. 9). In both vagus-intact and vagotomized conditions, a large phase advance was elicited by light pulses occurring during inspiration, consistent with our results in vitro. Light pulses during the expiratory phase elicited a phase delay, with longer delays near the end of Te. However, in vagotomized mice, light pulses had little effect during the time

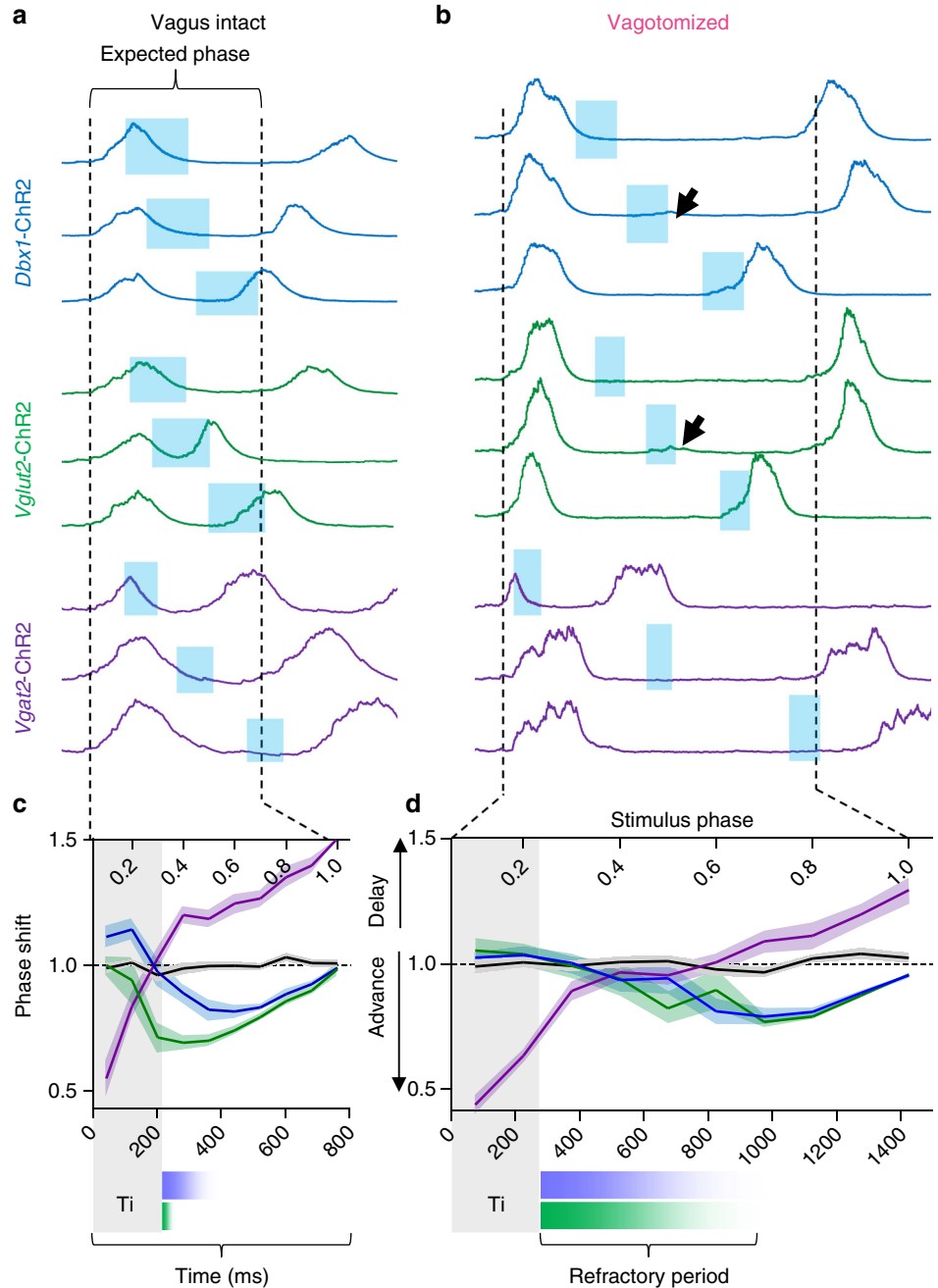

**Fig. 9** Vagal sensory feedback limits the refractory period for excitatory preBötC populations. **a, b** Example integrated XII nerve activity during brief (100–200 ms) light stimulation at different time points within the respiratory cycle in vagus-intact (**a**) and vagotomized (**b**) Dbx1-ChR2, Vglut2-ChR2, and Vgat-ChR2 mice. Note the difference in the refractory period for partially overlapping Vglut and Dbx1 populations in vagus-intact mice. Arrows indicate failed bursts near the end of the refractory period. Traces are scaled to the expected duration of the respiratory cycle (expected phase). **c, d** Quantified phase shift elicited during stimulations across the respiratory cycle (stimulus phase 0 = beginning of inspiration) in vagus-intact (**c**) (Dbx1, $n = 10$; Vglut2, $n = 10$; Vgat, $n = 5$; Cre−, $n = 9$) and vagotomized mice (**d**) (Dbx1, $n = 9$; Vglut2, $n = 7$; Vgat, $n = 3$; Cre−, $n = 9$; mean ± s.e.m.). Data from Cre− control mice shown in black. Refractory periods for Dbx1-ChR2 (blue) and Vglut2-ChR2 (green) shown on equivalent time scales relative to the average respiratory cycle duration in vagus-intact and vagotomized conditions. Note the exaggerated refractory period in vagotomized mice and the minimal effect of Vgat-ChR2 stimulation during the refractory period

following inspiration that corresponded to the refractory period (Note the flat area of the curve in Fig. 9d), suggesting that activity of inhibitory neurons during the refractory period has minimal effects on breathing (also see Fig. 5d). Interestingly, in contrast to in vitro preparations, stimulation of inhibitory neurons during expiration did not evoke postinhibitory rebound in vivo[23], calling into question models of respiratory rhythm generation

implicating postinhibitory rebound as a driver of rapid breathing based on a very slow non-physiological rhythms generated in vitro[40].

Finally, we optogenetically stimulated inhibitory preBötC neurons threshold-triggered by integrated XII nerve activity. If the preBötC was stimulated unilaterally, breathing frequency was increased initially but then became irregular in both vagus-intact

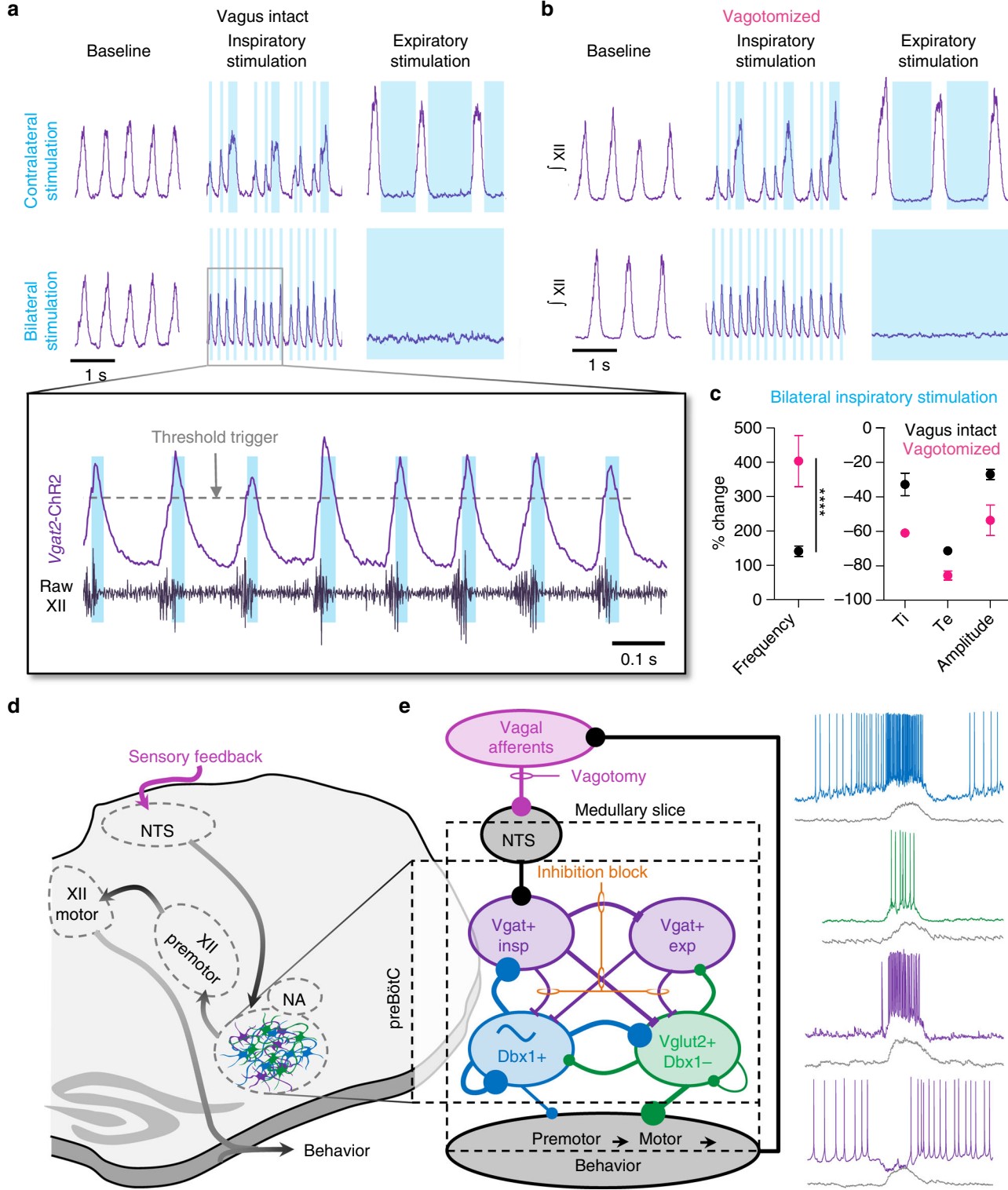

**Fig. 10** Rapid breathing is elicited by phasic inhibition during inspiration. **a**, **b** Example XII activity at baseline and during unilateral or bilateral light stimulation during the inspiratory or expiratory phase in vagus-intact (**a**) and vagotomized (**b**) *Vgat*-ChR2 mice. Expanded view demonstrates precise timing of light pulses triggered by the onset of inspiration. Note the irregular rhythm elicited by contralateral stimulation. **c** Quantified changes in frequency, Ti, Te, and XII amplitude during bilateral inspiratory light stimulation in vagus-intact ($n = 5$; black) and vagotomized ($n = 3$; pink) mice (mean ± s.e.m.; two-way ANOVA and Bonferonni's multiple comparisons test; ****$p < 0.0001$). **d**, **e** Model of interactions between excitatory and inhibitory preBötC populations controlling breathing frequency. Representation of the heterogeneous organization of the preBötC and simplified pathway from sensory input to rhythmogenesis to behavioral output (**d**). Hypothesized interactions between preBötC subpopulations (**e**). Line weight indicates the relative strength of connections. Interactions determine the activity pattern of each neuron type (right)

and vagotomized mice (Fig. 10a, b), consistent with our observations in vitro. Since the irregularity could reflect discoordination between the left and right preBötC due to far fewer commissural connections between inhibitory versus excitatory neurons[48], we stimulated the preBötC bilaterally during inspiration. Bilateral inspiratory stimulation elicited a stable increase in breathing frequency that was larger in vagotomized (403.3 ± 74.2%) versus vagus-intact (140.9 ± 15.4%) mice (Fig. 10c). Increased frequency was due to decreases in both Ti and Te. Importantly, unlike stimulation of excitatory preBötC neurons (see Fig. 8), inspiratory stimulation in vagotomized *Vgat-ChR2* mice increased breathing frequency to 4.6 ± 1.0 Hz, far beyond spontaneous frequencies with the vagus intact (1.9 ± 0.3 Hz) and beyond the maximum frequency predicted by the Dbx1 refractory period (~2.5 Hz in vagus-intact and ~1.1 Hz in vagotomized mice). Unilateral stimulation of inhibitory neurons during expiration decreased breathing frequency, whereas bilateral expiratory stimulation caused apnea[23] (Fig. 10a, b). Collectively, these data reveal a critical role for inhibition in the generation of high-frequency breathing in vitro and in vivo.

## Discussion

Unraveling the role of inhibition in the generation of rhythmic activity has been the focus of many investigations. Beginning with the concept that reciprocal inhibition plays a critical role in rhythm generation[49], many studies have demonstrated that synaptic inhibition is present in most rhythmogenic networks[3,9,50–53]. The respiratory network is no exception, and various computational models assume that inhibitory mechanisms are essential for rhythmogenesis[54–56]. It came as a great surprise when numerous studies revealed that network rhythmicity persists after blockade of synaptic inhibition[44,57–60] (Fig. 3b). These findings were initially dismissed, since early experiments were conducted in vitro. But, although still debated[21], rhythmicity persists even when synaptic inhibition is blocked in vivo[20] (Fig. 6b). The preBötC contains a large portion of rhythmically active inhibitory neurons (~37%; Fig. 4b), some of which may even possess autonomous bursting properties[18]. However, these neurons cannot generate the inspiratory rhythm, as they would inhibit activity during inspiration[18]. Thus the presence of inhibition in the respiratory network poses an unresolved puzzle: Not only does rhythmicity persist without inhibition, but what is the benefit of inhibitory mechanisms if they oppose rhythmogenesis[61]?

PreBötC burst-generating mechanisms are transiently suppressed following the transition from inspiration to expiration, during a phase termed post-inspiration[29,62] (Fig. 2a). A similar refractory period has been observed following network bursts in hippocampal[63,64], cortical[65], and spinal cord cultures[66]. It has been proposed that depletion of synaptic vesicles at excitatory synapses results in burst termination[29,67] and may also contribute to the refractory period[29]. However, it is possible that distinct mechanisms underlie these processes[66]. Our data support an alternative mechanism whereby an intrinsic membrane property of preBötC neurons underlies the refractory period. Many preBötC neurons become hyperpolarized following inspiratory bursts[27,36] (Fig. 2c–e). The specific ionic conductance leading to this AHP is unknown, but activity-dependent outward currents, e.g., the $Na^+$-dependent $K^+$ current (INa-K), $Na^+/K^+$ ATPase electrogenic pump current (Ipump), ATP-dependent $K^+$ current (IK-ATP), and calcium-activated potassium channels[57,68–70] have been identified in preBötC neurons that could contribute. Indeed, this AHP is proportional to the preceding depolarizing burst drive potential (Fig. 2c). Therefore, more excitable neurons are predicted to have a longer refractory period. Moreover, in a

network thought to rely on recurrent excitation for burst initiation[71,72], reduced excitability during the refractory period likely delays the onset of excitatory interactions percolating through the network, thereby increasing the period between bursts. Indeed, we found that longer refractory periods were associated with slower breathing frequencies (Figs. 3d and 9), and increased activity of excitatory neurons during inspiration decreases breathing frequency by increasing expiratory time (Fig. 8a). Thus excitability of excitatory preBötC neurons must be controlled to prevent hyperactivity and long refractory times.

Inhibitory neurons active during inspiratory bursts are prevalent within the preBötC[17–19] (Fig. 4b), and inhibition within the preBötC is further facilitated by vagal sensory feedback in response to lung stretch during inspiration[20] (Fig. 6, Supplementary Fig. 8). Concurrent inhibition of preBötC excitatory neurons critically regulates network synchrony[61,73], limiting overall activity and the refractory period (Fig. 3), and is therefore a powerful modulator of breathing frequency. Indeed, increasing the activity of inhibitory neurons phasically during inspiration increases frequency both in vitro (Fig. 4b) and in vivo (Fig. 10a, b). Conversely, removing inhibition increases activity during inspiratory bursts, prolongs the refractory period, and decreases burst frequency in the preBötC (Fig. 3). Most previous studies have inferred frequency changes during blockade of inhibition by measuring XII motor output[74]. Note, however, that transmission to XII depends on the excitability of the preBötC, which will be altered by blockade of inhibition and may therefore skew the interpretation of frequency changes[59]. Both GABAergic[17] and glycinergic[23] mechanisms seem to contribute to modulation of the refractory period and breathing frequency in the preBötC (Fig. 3); however, whether these are distinct groups of neurons or a single population that co-expresses glycine and GABA remains to be determined.

In vivo, removal of sensory feedback inhibition prolongs the refractory period (Fig. 9) and increases XII amplitude with a proportional decrease in breathing frequency and flexibility[20,28,46] (Fig. 6b–e). Although mechanisms downstream of the rhythm-generating circuit can further shape respiratory motor outputs, we find that hyperexcitability of excitatory preBötC neurons results in a slow and large amplitude breathing pattern. Indeed, a similar effect was observed following specific blockade of inhibition within the preBötC. This effect is not observed following removal of inhibition in computational models of the respiratory network[75]. Further, network models replicate the prolongation of Ti during simulated vagotomy[55,76]; however, most fail to reproduce the large increase in Te that dominates the effect of vagotomy on breathing frequency in vivo, highlighting the importance of incorporating principles of concurrent inhibition and refractory dynamics into models of rhythm generation.

Neurons that are active out of phase with inspiratory bursts (i.e., have expiratory activity) are estimated to comprise only ~10% of preBötC neurons[41]. However, we found that most (if not all) expiratory preBötC neurons are inhibitory (Fig. 4c). While inhibitory, inspiratory neurons promote rapid breathing, inhibitory, expiratory neurons slow breathing (Fig. 4d). Thus different populations of inhibitory neurons have competing roles in regulating frequency[44]. Our data suggest that inhibitory neurons exhibit an expiratory phenotype because they receive more input from inhibitory than excitatory neurons active during inspiration (Figs. 4c and 10e). Indeed, much like following the pharmacological blockade of inhibition[42,43] (Fig. 4c), expiratory neurons discharge in phase with inspiration during severe hypoxia[77], which also reduces inhibition within the preBötC[37]. Moreover, during lung stretch, which we predict increases the activity of inspiratory inhibitory neurons (Fig. 10e), expiratory neurons are

hyperpolarized[78]. This inhibitory feedback mechanism may stabilize rhythmogenesis by synchronizing excitatory neurons via mechanisms of postinhibitory rebound. In future modeling studies, it will be interesting to test this network structure on the specific stability and flexibility characteristics of the preBötC rhythm. However, our results suggest that postinhibitory rebound is unlikely to be a driver of rapid breathing[40] under physiological conditions, since rebound from inhibition does not overcome the refractory period (Fig. 5d) and does not generate inspiratory bursts in vivo[23] (Fig. 9).

Excitatory preBötC neurons have varied activities that have been grouped into distinct rhythm- and pattern-generating microcircuits. Rhythmogenic neurons with augmenting pre-inspiration activity are primarily Dbx1 derived, while pattern-generating neurons are excitatory somatostatin expressing or Dbx1 derived[28,34]. In this study, we compared the effects of stimulating partially overlapping Dbx1+ and Vglut2+ preBötC subpopulations[24,26] on respiratory frequency and pattern (amplitude). Although stimulation of Dbx1+ and Vglut2+ populations had indistinguishable effects in vitro (Fig. 2b, Supplementary Fig. 1a) and in vagotomized mice in vivo (Figs. 7b, 8b, and 9b), with vagal sensory feedback intact, the refractory period for Vglut2-ChR2 was reduced relative to Dbx1-ChR2 (Fig. 9a). It is possible that Vglut2-ChR2 stimulation leads to greater excitation of the same subpopulation of neurons due to activation of Vglut2+ terminals innervating the preBötC. Although this may be a possibility, it is difficult to reconcile how the differential effect could be limited to the in vivo vagus-intact condition, to changes in burst frequency versus amplitude (Fig. 7c), and to the expiratory versus inspiratory phase (Fig. 8c). Because some preBötC neurons are inhibited during lung inflation while others are excited[78], we speculate that vagal feedback increases the activity of inhibitory, inspiratory preBötC neurons[79,80] and that these neurons preferentially inhibit non-Dbx1 excitatory neurons, reducing their activity relative to rhythmogenic Dbx1 neurons (working model displayed in Fig. 10d, e). Since the refractory period is determined by neuronal activity (Fig. 2c–e), non-Dbx1 excitatory neurons would be expected to have a shorter refractory period than Dbx1 neurons in the presence of sensory feedback inhibition. Although this working model of the preBötC best fits our results, the proposed connectivity lacks direct evidence, which will be an important avenue for future investigations.

In summary, our findings support a conceptually novel, unifying hypothesis: "the Yin and Yang of rhythm generation", in which both excitatory and inhibitory neurons are required to allow neuronal oscillators to generate rhythmicity over a wide dynamic frequency range. While excitatory mechanisms are both necessary and sufficient for respiratory rhythmogenesis, the resulting rhythm is slow and inflexible. By modulating the refractory time, interactions between excitatory and inhibitory neurons allow the preBötC to generate physiological breathing frequencies and endow it with the flexibility to adapt to environmental, metabolic, and behavioral needs that are essential for survival. This interdependence between inhibition and excitation may be a principle that applies to other rhythm-generating networks as well.

## Methods

**Animals.** Neonatal (P4–P12) and adult (P47–P380) male and female C57BL/6 mice bred at Seattle Children's Research Institute were used for all experiments. *Vglut2-ires-Cre* and *Vgat-ires-Cre* homozygous breeder lines were obtained from Jackson Laboratories (Stock numbers 028863 and 016962, respectively). Heterozygous *Dbx1^{CreERT2}* mice were donated by Dr. Del Negro (College of William and Mary, VA) and a homozygous breeder line was generated at SCRI. Cre mice were crossed with homozygous mice containing a floxed STOP channelrhodopsin2 fused to a tdTomato (Ai27) or EYFP (Ai32) reporter sequence donated by Dr. Hongkui Zeng (Allen Brain Institute, WA). Mice were randomly selected for experiments

from the resulting litters by the investigators. A subset of Cre mice were crossed with homozygous Cre-dependent enhanced green fluorescent protein (ZsGreen1) expressing mice (Ai6) for visualization of cell bodies (Jackson Laboratories stock number 007906). *Dbx1^{CreERT2}* dams were plug checked and injected at E10.5 with tamoxifen (24 mg/kg, intraperitoneal (i.p.)) to target preBötC neurons[24,26]. Dbx1 neurons may have functional variation depending on their "birth date" (i.e., timing of tamoxifen administration); however, this has not been specifically studied. Offspring were group housed with ad libitum access to food and water in a temperature-controlled (22 ± 1 °C) facility with a 12 h light/dark cycle. All experiments and animal procedures were approved by the Seattle Children's Research Institute's animal care and use committee and were conducted in accordance with the National Institutes of Health guidelines.

**In vitro experiments.** Transverse brainstem slices containing the preBötC were prepared from postnatal day 4–12 mice. In brief, the brainstem was isolated and glued to an agar block (dorsal surface to agar with the rostral end up) and submerged in artificial cerebrospinal fluid (aCSF; in mM: 118 NaCl, 3.0 KCl, 25 NaHCO₃, 1 NaH₂PO₄, 1.0 MgCl₂, 1.5 CaCl₂, 30 D-glucose) equilibrated with carbogen (95% O₂, 5% CO₂). aCSF had an osmolarity of 305–312 mOSM and a pH of 7.40–7.45 when equilibrated with gas mixtures containing 5% CO₂ at ambient pressure. Serial 200 μm sections were made through the medulla, and a single 560 μm slice containing the preBötC was retained based on anatomical landmarks such as the XII nucleus, inferior olive, and size of the fourth ventricle. Rhythmic activity from the preBötC was induced by raising extracellular KCl to 8.0 mM. Extracellular population activity was recorded by positioning a glass pipette (tip resistance <1 MΩ) filled with aCSF on the surface of the slice over the preBötC. Signals were amplified 10,000×, filtered (low pass, 300 Hz; high pass, 5 kHz), rectified, integrated, and digitized (Digidata 1550 A, Axon Instruments). Intracellular recordings were made from preBötC neurons using the blind patch clamp approach with a multiclamp amplifier in current clamp configuration (Molecular Devices, Sunnyvale, CA). Recording electrodes were pulled from borosilicate glass (4–8 MΩ) using a P-97 Flaming/Brown micropipette puller (Sutter Instrument Co., Novato, CA) and filled with intracellular patch electrode solution containing (in mM): 140 potassium gluconate, 1 CaCl₂, 10 EGTA, 2 MgCl₂, 4 Na₂ATP, and 10 Hepes (pH 7.2). Extracellular and intracellular signals were acquired in the pCLAMP software (Molecular Devices, Sunnyvale, CA). A glass fiber optic (200 μm diameter) connected to a blue (447 nm) laser and DPSS driver was positioned above the preBötC contralateral to the extracellular electrode and ipsilateral to the intracellular electrode. In experiments where bilateral stimulation was performed, a second laser coupled to 200 μm fiber optic was used and triggered simultaneously with the same TTL signal. Laser power was set <150 μW. In some experiments, fast synaptic inhibition was blocked by bath application of strychnine (1 μM) and gabazine (1 μM). Excitatory Dbx1 neurons were identified optogenetically following pharmacological isolation by bath application of a cocktail of synaptic blockers (CNQX, 20 μM; CPP, 10 μM; strychnine, 1 μM; gabazine, 1 μM).

**In vivo experiments.** Adult mice were anesthetized with urethane (1.5 mg/kg, i.p.) and placed supine on a custom surgical table. The trachea was exposed through a midline incision and cannulated caudal to the larynx with a curved (180 degree) tracheal tube (24 G). Mice were then allowed to spontaneously breathe 100% O₂ throughout the remainder of the surgery and experimental protocol. Electrocardiogram leads were placed on the fore and hind paw to monitor heart rate. Core temperature was monitored and maintained with a heat lamp. Adequate depth of anesthesia was determined via heart rate and breathing frequency responses to toe pinch and adjusted if necessary with supplemental urethane (i.p.) prior to experimental protocols. The hypoglossal nerve (XII) was isolated unilaterally, cut distally, and recorded from using a fire-polished pulled glass pipette filled with aCSF connected to a suction electrode. The vagus nerve (X) was isolated bilaterally, but left intact until a specific point in the experimental protocols (see below). The trachea and esophagus were removed rostral to the tracheal tube, and the underlying muscles were removed to expose the basal surface of the occipital bone. The portion of the occipital bone and dura overlying the ventral medullary surface were removed with microscissors. The brainstem surface was perfused with warmed (~36 °C) aCSF equilibrated with carbogen (95% O₂, 5% CO₂). XII electrical activity was amplified (10,000×), filtered (low pass 300 Hz, high pass 5 kHz), rectified, integrated, and digitized (Digidata 1550 A, Axon Instruments). A glass fiber optic (200 μm diameter) connected to a blue (447 nm) laser and DPSS driver was placed in light contact with the brainstem (contralateral to XII recording) and positioned for maximal effect on breathing frequency during light stimulation. The basilar artery, caudal cerebellar artery, and vertebral arteries were suitable landmarks for the X and Y positioning of the fiber optic. The most rostral XII nerve rootlet also served as an effective rostrocaudal landmark, with the fiber optic typically positioned immediately rostral to this rootlet. In bilateral stimulation experiments, a second fiber and laser were used and triggered simultaneously with the same TTL signal.

**Stimulation protocols.** Three types of stimulation paradigms were used: (1) For continuous stimulation, inspiratory activity was recorded in 30 s sweeps containing a 10 s continuous light stimulus (2–10 sweeps for each stimulation). The

percentage of change in instantaneous frequency and amplitude of each XII burst relative to the averaged values during baseline was calculated, binned in 1 s intervals, and plotted over time. To examine laser-power-dependent effects on breathing, light intensity was varied from 0 to 13 mW. However, non-specific effects were observed in Cre− control mice >7 mW. Therefore, laser power was set at <7 mW for all other in vivo stimulations. (2) For phase-dependent stimulation, integrated XII nerve activity was thresholded in order to trigger a TTL pulse during the rising phase or the falling phase of the integrated XII signal (Clampex Software), thereby isolating the laser stimulus to the inspiratory or expiratory respiratory phase. (3) For randomized brief stimulations, short 100–200 ms light pulses were delivered at 10 s intervals to stimulate the preBötC at random time points during the respiratory cycle (30–150 stimulations/experiment). The respiratory cycle was defined as the onset of inspiration to the onset of the subsequent inspiration. The phase shift elicited by each stimulation was calculated as the duration of the respiratory cycle containing the stimulus, divided by the average of the two preceding respiratory cycles (expected phase). The phase of preBötC stimulation was calculated as the time between the onset of inspiration and the stimulus onset, divided by the expected phase. The average phase shift was then plotted against the stimulus phase in bins containing 1/10 of the expected phase.

Each stimulation paradigm was performed during fast, eupneic breathing with the vagus intact and during slow breathing following vagotomy. XII nerve activity was continuously recorded while the vagus nerves were cut bilaterally. In a subset of vagus-intact mice, pulled glass injection pipettes were used to inject (150 nl) a mixture of strychnine (250 µM) and gabazine (250 µM) diluted in aCSF bilaterally into the preBötC to block synaptic inhibition. Thirty to 60 s of XII nerve activity before and 5–10 min after vagotomy or injection of strychnine and gabazine were used to quantify changes in breathing parameters.

Following electrophysiological experiments, the position of the fiber optic on the ventral surface of the medulla was marked with an injection of fluorescent beads (Fluorospheres). Fresh wet or fixed and cryoprotected (see below) whole brainstems were imaged, and the coordinates of the stimulation site were measured relative to the midline ($X$ direction) and both the most rostral point of the caudal cerebellar artery and the intersection of the vertebral arteries ($Y$ direction). Distances were normalized to the half-width of the brainstem at the level of the caudal cerebellar artery to correct for variations in animal size and differences due to tissue shrinkage from fixation in paraformaldehyde.

**Microscopy and immunohistochemistry.** Following in vivo experimental protocols, adult mice were euthanized, the brainstem was removed, fixed overnight in paraformaldehyde (4% in phosphate-buffered saline (PBS)), cryoprotected (30% sucrose), and stored at 4 °C. The ventral surface of fresh or cyroprotected whole brainstems were imaged (Olympus SZX16) and rostrocaudal and mediolateral coordinates of the stimulation site were measured (Olympus, cellSens software) relative to landmarks on the ventral medullary surface (Fig. 6f).

A subset of Cre-expressing mice were crossed with Cre-dependent ZsGreen (Ai6) mice. Fixed and cryoprotected brainstems were embedded and frozen in OCT (−80 °C) and sectioned (40 µm) in the transverse plane. Floating sections containing the preBötC were rinsed in PBST (0.1% Triton X-100 in 1× PBS, 20 min at room temperature), washed in block solution (10% donkey serum in PBST, 3 times, 20 min at room temperature), and incubated overnight at 4 °C in goat anti-ChAT primary antibody (1:100, Millipore, catalog Ab144p, lot 2500408) diluted in blocking solution. Sections were then washed in PBS (3 times, 20 min at room temperature) and incubated in donkey anti-goat secondary antibody at room temperature for 2 h (1:250, Alexa Fluor 568, Invitrogen, catalog A11057, lot 1235787). Sections were rinsed in block solution (3 times, 20 min at room temperature), incubated in 4,6-diamidino-2-phenylindole for 2 min, washed in PBS (5 min), and mounted on slides with Flouromount.

Half brainstem fluorescent images were acquired on a Leica DM 4000 B epifluorescence microscope equipped with 405, 488, and 548 nm laser lines and a Leica 2.5× objective (416ed, 506083). Inset images were acquired on a Zeiss LSM 710 laser scanning confocal microscope equipped with 405, 488, and 548 nm laser lines and a Zeiss 20× objective (420650-9901). Confocal images (20×) are z-projected stacks (summed intensity of all planes). All images were post-processed using the Image-J software (Version 1.48).

**Statistical analysis.** Statistical analyses were performed using the GraphPad Prism 5 software. Data distributions were tested with Shapiro–Wilk normality test. Unless otherwise noted, normal data were compared using an appropriate one-way or two-way analysis of variance with Bonferonni's multiple comparisons tests. Data failing the Shapiro–Wilk normality test were compared using Friedman's test and Dunn's multiple comparisons test. Variance was similar between groups unless otherwise noted. Welch's correction was used for unequal variances. Differences were considered significant at $p < 0.05$. Investigators were not blinded during analysis. Sample sizes were chosen on the basis of previous studies.

**Data availability.** The authors declare that data supporting the findings of this study are available within the paper and available upon request from the corresponding author.

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

## Acknowledgements

We thank Dr. Christopher Del Negro for donation of *Dbx1*$^{CreERT2}$ mice and NIH (grant nos. R01 HL126523, P01 HL 090554 and F32 HL134207) for funding these projects.

## Author contributions

Conceptualization, N.A.B. and J.M.R.; methodology, N.A.B. and J.M.R.; investigation, N.A.B., H.C.B.; formal analysis, N.A.B.; writing—original draft, N.A.B.; writing—review and editing, N.A.B. H.C.B. and J.M.R.; funding acquisition, N.A.B. and J.M.R.; visualization, N.A.B.; supervision, J.M.R.

## Additional information

**Competing interests:** The authors declare no competing financial interests.

