## [Peer Review File · Nature Communications]

Reviewers' comments:

Reviewer #1 (Remarks to the Author):

Baertsch et al study on the "Interdependence of excitation and inhibition in respiratory rhythm generation" deals with the so far puzzling problem of the role played by the large population of inhibitory neurons in the preBötzinger complex. This structure provides the core circuit for respiratory pattern generation. It had been shown earlier that respiratory bursting remains after a blockade of all inhibition, showing that excitatory neurons are sufficient to generate respiratory activity. They combine slice and in vivo work in mice in which ChR had been introduced in either the inhibitory neurons or in the excitatory Dbx-subpopulation or finally in all excitatory neurons in preBötzinger (PB). They show that the inhibitory neurons actually play an important role in regulating the burst frequency and enhancing the frequency range of bursting.

When all inhibition has been blocked, the respiratory bursts increase in amplitude and a slow rate with a long "refractory" phase. Similarly for individual excitatory neurons, if activated by light pulses, they express a long afterhyperpolarisation, which increases markedly with the increase in the preceding light-induced burst of action potentials. If instead the population of inhibitory neurons is activated by pulses of light during a respiratory burst, the burst amplitude will decrease, and the refractory period, following the burst will also shorten and thereby the burst frequency will be increased. When inhibitory neurons are activated during the interval to the next burst, they will as expected instead delay the subsequent burst.

The inference of the burst frequency increase with inhibition during the burst is presumably that the level of activity in the population of excitatory neurons is decreased, and thereby a shorter afterhyperpolarisation (AHP) will result in the entire population of interacting excitatory neurons. Activation of the vagal input (lung stretch receptors) also results in a shorter cycle length presumably through an action on inhibitory neurons. The paper contains a wealth of important information, is well written and the figures to the point.

Major comments:

The abstract could better convey the results and the authors interpretation. Rather than mentioning "refractory dynamics", I think the straightforward explanation that the AHP of single excitatory interneurons is markedly dependent on the degree of activation (number and frequency of action potentials) of single excitatory interneurons (Fig 1H, I), and the population as well. Regulating the degree of PB excitatory interneuron activity by the inhibition will automatically regulate the duration of the AHP in the population of cells.

One problem, perhaps handled at a downstream level, is that whereas the level of PB activity decreases with increased burst frequency, the net output to the respiratory muscles during inspiration increases with respiratory rate. Please comment!

The results in Fig 2C (page 8 and also 10 (Fig 3)) showing the effect of light pulse activation

of Dbx neurons during the refractory period are to the point, but it would be more informative to plot them as phase-response curves (plotted in relation to an unperturbed cycle). In this context it would also be important to report the exact timing of the inhibitory light pulse within different parts of the PB-burst. Whether the effect will differ in the initial, middle or late part of the PB-burst – could be clearly represented in a phase response curve. As well as the effects during the refractory period.

The legends are in general too brief – often lack of sufficient information to critically evaluate the figures – for instance legend Fig 4. In Fig 4 E, one has to guess which bar is what, and the meaning of 4H is not obvious. What is CPAP? The manuscript and figures should be understandable for a general neuroscience readership particularly in this journal

Page 12-13 Effect of vagal input.

The authors should refer to the cat data of Clark and Euler 1972 showing that vagotomy causes a marked slowing of the respiratory burst rate to a constant slow rate rate. Varying the CO₂ drive changed the amplitude at the output level (phrenic nerve) but did not affect the burst frequency. Also consider this in relation to the current results.

Minor comments:

Page 9 lines 177-184

Figure S2A shows a neuron and population recordings for what is referred to as a fictive sigh, which has a somewhat longer burst interval and a larger afterhyperpolarisation. It would seem important to define the criteria better for what is considered as a fictive sigh, rather than what seems just an unusually large burst followed by a longer refractory phase. If this cannot be done – this part could be omitted.

Page 13, lines 270 to 280,

This would seem to fit better in Methods.

Page 20 line 431

Why “limiting synchronization” rather than just increased overall activity in the PB population – due to excitatory interaction among the excitatory PB neurons. The higher the level of activity in each neuron – the larger AHP and refractory time.

Page 20, lines 437-439.

Should Cheyne-Stokes breathing be relevant in this context?

Page 21 lines 452 to 461

Sr has an additional effect in that it blocks the Ca²⁺ site internally on Kca channels, which could explain the findings

Reviewer #2 (Remarks to the Author):

I read this manuscript three times to prepare this review because it contains a great deal of data and covers a lot of ground. The manuscript addresses important contemporary problems that pertain to central pattern generation as well as sensorimotor integration. It could make important contributions to the field(s) of CPGs and respiratory neurobiology. However, I think there are a few interpretive problems that must be dealt with as well as many other items that require clarification.

Addressing the role of GABAergic synaptic inhibition in respiratory rhythm, the authors demonstrate that fast inhibition can operate on a cycle-to-cycle basis to modulate the amplitude and frequency of respiratory rhythm. Modulation of respiratory (breathing) frequency is the main focus; the authors compare to what extent inhibitory vs. excitatory interneurons can influence frequency. They find that inhibitory mechanisms are more powerful, which is a new insight of general interest.

The effect of fast synaptic inhibition on the amplitude of breaths has been demonstrated before for glycinergic neurons (Sherman et al. Nat Neurosci 18: 408-414, 2015) but the authors here address GABAergic neurons. It is possible that these are the same or overlapping populations that co-express GABA and glycine. I recommend that the authors comment on that in Discussion.

As for frequency, Sherman showed that activation of glycinergic preBotC neurons during inspiration attenuates the inspiratory breath and phase advances the core oscillator. Those authors applied single brief pulses to Glyt2 mice injected with AAV to express ChR in the preBotC and constructed the phase-response curve. The present manuscript uses Vgat2-ChR2 transgenic mice and applies a more sophisticated approach by triggering light pulses off of the inspiratory waveform for repetitive cycle-triggered photo-stimulation. Doing so, they show that phasic inhibition very potently modulates frequency and enables the 2-3 Hz breathing frequencies characteristic in vivo. Interestingly, just photo-stimulating excitatory neurons is not enough; the frequency only increases a little bit. So, the results are an advance compared to Sherman, although not totally novel. Again, commenting on the potential overlap between GABA and glycine preBotC neurons would be helpful so the reader can interpret the present study and the Sherman study in context (are they the same or overlapping populations, or separate populations that accomplish similar tasks? Are GABA and glycine preBotC interneurons redundant?)

Another part of the manuscript looks at intrinsic refractory mechanisms in the preBotC. The PI (Ramirez) has carved out an interesting problem, that of post-inspiratory mechanisms. Recently his group identified the "post-inspiratory complex" (PiCo). This paper follows that line of inquiry. They show that synaptic inhibition influences the preBotC refractory period and thus breathing frequency; when inhibition is blocked, the refractory period is enhanced, frequency is correspondingly diminished.

The phase-response curves in Fig. 7Bi and Bii are the best part of the paper (it's a shame they come so late). The steep purple curve in the vagus intact mouse shows how sensitive

the phase (frequency) is to inhibition. That curve gets quite flat during the refractory period in the vagotomized mouse. This is evidence that supports the claim that sensory feedback from lung mechanoreceptors, i.e., one major source of inhibition, does influence burst termination. When that feedback inhibition is removed (by vagotomy) the preBotC neurons generate larger bursts, making them refractory either by synaptic depression or intrinsic activity dependent outward currents, and thus become unresponsive during that refractory period, hence the flat phase-response curve after vagotomy (in response to Vgat2-ChR2 photo-stimulation).

My enthusiasm for the paper notwithstanding, I do have several problems that should be addressed:

The block of inhibition in previous slice studies did NOT modulate XII/preBotC burst frequency. That result has been widely replicated. A particularly comprehensive report by Gray et al. Science (1999) shows that a variety of peptides modulate respiratory frequency, but that fast chloride-mediated inhibition does not. And, they showed that peptide modulation of the rhythm stays the same whether fast inhibition is present or not (see Fig 1B in Gray et al Science 286:1566, 1999). There are many other studies from the Smith lab, Ballanyi (Brockhaus & Ballanyi), and Greer (Ren & Greer). Why are these present results so different from past studies?

The authors use two transgenic Cre- driver mice for in vivo studies. One is linked to Dbx1 and the other is linked to Vglut2. In the preBötC these Cre- drivers should target largely the same population of neurons, but outside the preBötC there will be many more Vglut2 excitatory neurons than just those derived from Dbx1-expressing precursors. Furthermore, outside preBötC, Dbx1-derived neurons can be excitatory or inhibitory. Therefore, in the optogenetic protocols in Figure 7, where photo-stimulation of Vglut2-ChR2 evokes bursts in the refractory period of Dbx1 neurons, the effect is quite credibly explained by a plethora of ChR2-expressing Vglut2 synaptic terminals in the preBötC from cell bodies located elsewhere. There is NO reason why one would suspect a latent population of non-Dbx1 excitatory neurons in the preBötC. The results could be explained simply by activating a lot of Vglut2 terminals, which is sufficient to overcome the refractory state and evoke a burst. Acknowledging this alternative explanation is absolutely critical, and then adjusting the interpretation and conclusion accordingly.

A lesser concern: the authors should consider that photo-stimulation in Vglut2-ChR2 mice will activate only neurons whereas in Dbx1-ChR2 mice the photo-stimulation protocol will also activate Dbx1-derived glia, which could be 30-40% of the cells undergoing Cre-mediated recombination. There could be purinergic effects related to gliotransmission and perhaps some other influences not yet widely understood.

The authors frame their conclusions in terms of excitatory and inhibitory mechanisms that control rhythmogenesis. I recommend they distinguish rhythm generation (due to excitatory synaptic mechanisms) from its regulation (due to inhibitory synaptic mechanisms). The two are more easily understood, in my view, once thus distinguished.

SPECIFIC MAJOR CONCERNS

Line 34: Excitation and inhibition are not interdependent for respiratory rhythm generation. Synaptic excitation is obligatory for rhythmogenesis, whereas synaptic inhibition is dispensable. The line in the Abstract as written is misleading. Inhibition is relevant for modulating the timing and the magnitude of inspiratory bursts, which influences the rhythm but that does not mean that they are "interdependent for respiratory rhythm generation".

Lines 128-131: the cell in Fig. 1G hyperpolarizes a lot (>5 mV) during post-inspiration, which is a characteristic presciently identified by Rekling et al. *J Neurophysiol* 75: 795-810, 1996. However, Rekling et al. pointed out that these so-called Type 1 preBotC neurons were approximately half of the inspiratory neurons they recorded. Picardo et al. (2013) working specifically on Dbx1 preBotC neurons obtained largely the same data and came to a similar conclusion that the preBotC neurons with large post-burst after-hyperpolarizations (AHPs), the so-called Type 1 neurons, were about half of the Dbx1 preBotC population. My question: Do the authors find $\sim 50\%$ of their recorded Dbx1 neurons like Fig. 1G with the deep post-inspiratory AHPs or is their fraction higher (or lower). And, should that be acknowledged in Results?

Lines 208-209: The optogenetic pacing protocol here is a clever demonstration that inhibition can influence frequency. Nevertheless, the experiment itself shows the same qualitative point as Sherman et al. *Nat Neurosci* 18(3): 408, 2015, who showed that optogenetic activation of glycinergic (Glyt2, Slc6a5) preBotC neurons phase-advances the next breath. Here, by repeatedly applying photo-stimulation to Vgat+ preBotC neurons the authors just show that photo-stimulation phase advance works repeatedly and the rhythm speeds up as a result. This point largely recapitulates a point I made above in general comments.

Lines 214-218: Why don't the authors show these data regarding graded activation either in a main figure or supplemental?? I strongly encourage them to show those data.

Lines 218- 232 and Fig 3: I recommend that authors annotate the charts in Fig. 3G so they are a little easier to decipher on sight. The data are important but it is hard to know what all the measures and categories are just from the figure itself. Consider how the graphics can be improved so the reader does not have to stop and consult the legend

Lines 237-241: Did the authors ever patch-record a Vgat- neuron (in the Vgat-ChR2 mouse) while trying to evoke a preBotC burst by triggering post-inhibitory in the Vgat+ population? It would bolster the point they are making if they could show that the post-inhibitory rebound does NOT occur in the ~ 2 s refractory period. The field recording just shows that the preBotC burst does not occur during the ~ 2 s refractory period.

Lines 277-278: The authors write, "Extracellular recordings in this area confirmed inspiratory population activity that occurred prior to XII bursts". Can they show a trace to demonstrate that?

Lines 350-351: This is over interpretation. These mice are Vglut2-Cre;ChR2 mice, therefore all the excitatory axons of passage that enter in and synapse within the preBotC are evoked by light stimulation. The phase advanced inspiration in Vglut2-ChR2 mice (which does not occur in Dbx1-ChR2 mice) could be due to exciting so many excitatory ChR2-expressing synaptic terminals in the preBotC which then excite the core rhythm generators and overcome whatever refractory mechanisms are ordinarily preventing the light evoked burst in the ~200 ms refractory period in vivo. This largely recapitulates a general critique above.

Lines 462-466: I'm confused by the statement "We identified an alternative refractory mechanism by recording from synaptically isolated preBötC neurons; i.e. a membrane property that leads to hyperpolarization and reduced spiking following network bursts..." That refractory mechanism was first described by Reikling et al. J Neurophysiol 75: 795-810, 1996. It is not novel or alternative because it was earlier proposed to influence the respiratory rhythm generation mechanism.

Lines 473-474: Regarding the conclusion, "one would expect a period of reduced spiking during the refractory period to decrease burst frequency." Not necessarily. That presumes that recovery from refractoriness and burst initiation are linked. That may not be the case as synaptic vesicles restore and reset, and baseline membrane comes back to its baseline (cellular autonomous mechanisms), and then a process of percolation of excitatory synaptic activity initiates the next burst (network properties attributable to topology and number of interconnections among the constituent rhythm generators).

Lines 499-500: The authors conclude that stimulating Vglut2+ neurons evokes bursts during the Dbx1+ neurons' refractory period. However, it could be Vglut2+ synaptic terminals in the preBötC, whose cell bodies could be located anywhere, which evoke the burst. The vast majority of Vglut2 neurons in preBotC are Dbx1-derived, but there could be many more Vglut2 inputs to preBotC not Dbx1-derived. Again, this recaps a point in general comments.

Lines 502-503: In conjunction with the comment above, it is not possible on the basis of these data to conclude that non-Dbx1 excitatory interneurons can be rhythmogenic when the photo-stimulation effects, i.e., the ability to evoke bursts during the refractory period could be attributed to Vglut2+ terminals in the preBotC and NOT neurons located therein. This conclusion is not credible.

SPECIFIC MINOR CONCERNS

Lines 54-57: Certainly the preBotC serves as the "master clock" as the authors indicate, but its preeminent role is inspiratory rhythm generation. For the non-specialist, the "master clock" being introduced first might be misleading.

Lines 83-84: I recommend the authors also cite Kuwana et al. Eur J Neurosci 23: 667-74, 2006 because it characterizes GABAergic preBotC neurons, whereas Winter et al. (2009) deals with glycinergic neurons.

Lines 106-107: Tamoxifen-inducible Cre expression in Dbx1-CreERT2 mice, and its effect on fusion protein expression in the preBotC of intersectional offspring has very recently been quantified in detail by Kottick et al. *Physiol Rep* 5(11): e13300, 2017. I recommend the authors consider citing it.

Line 142: "The magnitude of reduced spiking..." is confusing. Consider rephrasing, e.g., "diminution of spiking ability"?

Lines 177-178: Why aren't the authors referring to some important in vitro 'sigh' work by their group Lieske et al. *Nat Neurosci* 3: 600-7, 2000, and others, Ruangkittisakul et al. *J Neurosci* 28: 2447-58, 2008?

Line 193: The 3D effect on the pie chart is superfluous ink (see: Tufte, ER. *Visual display of quantitative information*. Graphic Press: Cheshire, CT, 2001); i.e., the 3D effect does not enhance scientific communication. Also, the graphic itself is superfluous: having the fractions written in the text is enough, the pie chart does not make the point more clear in my view.

Lines 196-197: It is interesting that expiratory neurons became inspiratory (Fig. 3C here). Consider Shao & Feldman *J Neurophysiol* 77: 1853-60, 1997, who showed in slices that BIC or STR applied separately abolished expiratory activity in preBotC neurons but did not induce inspiratory modulation per se. On the other hand, zero-chloride ACSF made the XII nerve go tonic but seems to have made the expiratory neuron discharge what "seem" like bursts. Similarly, Brockhaus & Ballanyi *Eur J Neurosci* 10:3823-39, 1998, showed the expiratory-to-inspiratory transformation solely in response to BIC or STR. These precedents in the literature should be dealt with and might bolster their present result.

Lines 267-269: It's already well established that vagal afferents act on the preBötC via inhibitory mechanisms.

Line 273: recommend write "half", instead of the numeral fraction.

Lines 298-299: What are the non-specific effects here? I don't clearly see what they might be from Fig. S6.

Lines 374-375: This is an interesting observation (Fig. 7Bii) that activation of inhibitory neurons has little effect during the refractory period. That seems to be more evidence that the refractory period is attributable to synaptic depression or activity dependent outward currents in rhythmogenic preBotC interneurons.

Line 494: Regarding Dbx1 pattern-generating neurons in the preBötC, Cui et al. *Neuron* 2016 is an important citation, but I recommend also citing Wang et al. *eLife*, 2014, who demonstrated Dbx1 premotor neurons in the preBötC.

Line 561: The Dbx1 Cre-driver mouse line is described as tamoxifen inducible, with the "ERT2" appended to its nomenclature, in the Methods and earlier in the manuscript. The

acknowledgement lists it simply as Dbx1-Cre. I presume the mouse line is constant so I recommend the authors append ERT2 in the acknowledgement.

Reviewer #3 (Remarks to the Author):

The work of Feldman and colleagues supports the notion that the generation of the inspiratory oscillations in the preBoetzinger complex (preBotC) results from an all excitatory circuit. The latest incarnation of this work suggest that each excitatory cell tends to produce a subthreshold burst (burstlet) , and the next cycle of oscillatory output occurs when a critical number of burstlets occur at the same moment. This is reminiscent of oscillators in the inferior olive (IO).

Here, Ramirez show that inhibition is critical in setting the frequency of the oscillations, again reminiscent of what occurs in the IO. This result are extremely important as there were no prior data to suggest how the preBotzinger can oscillate over a very wide range of frequencies - easily 3 to 10 Hz in the rat. Further, the authors showed that oscillations at the sniffing frequency, synonymous with coordinated orofacial behavior, are completely dependent on inhibitory input.

Further, Ramirez and colleagues show that the mechanism is a control of excitation to reduce the refractory period, presumably by limiting the number of cells that are inactivated and drop out of producing burstlets. This is a very interesting control mechanism and one that has not been previously discussed in the CPG literature, which is dominated by data on oscillators with a fixed frequency.

Lastly, the authors assess the influence of the vagus nerve.

Suggestions to improve the manuscript.

1. "Rhythmicity is generated by neural networks through a combination of precisely timed excitatory and inhibitory mechanisms ..." I think this should be revised since the mechanism for rhythmicity in spike generation in the PreBotC and the IO is just excitation.
2. The data of Figures 1F and 2B implies that the authors can construct a phase resetting curve for (transhemispheric synaptic) excitation, i.e., the shift in period to the next burst relative to when the perturbation occurs relative to the unperturbed period (as estimated from previous cycles); like figure 7Bi, but with phase as the abscissa. This addition, presumably calculated from existing data, would be a valuable contribution for anyone hoping to model the system and possibly map it into a network of coupled (noisy) single-cell oscillators.
3. Similar to the above, the data of Figures 3H implies that the authors can construct a phase resetting curve for synaptic driven inhibition. They should do so.

In toto, the topic is important, the experiments are far from trivial, the data appear clean, and the logic of the study seems solid. I vote for publication.

Reviewer #4 (Remarks to the Author):

The manuscript titled "The interdependence of excitation and inhibition in respiratory rhythm generation" by Baertsch et al. is a series of well-designed and thoughtful experiments to determine the general function of inhibitory neurons in the breathing rhythm generator, the preBötC. The preBötC is composed of several thousand cells and half of these are inhibitory. However, because inhibitory neurons are not required for rhythm generation, their role in breathing has remained largely uncharacterized. Baertsch et al. use general transgenic lines to determine a role of inhibitory neurons in limiting the extent of depolarization of excitatory inspiratory neurons. The model predicts that the frequency of breathing is dictated by the refractory period of the preBötC following a burst and that inhibitory neurons can speed up the frequency of bursting by limiting the extent of excitation, thereby reducing the refractory period. Although the data and combination of in vitro and in vivo experiments are quite compelling, 4 primary concerns and 5 secondary concerns remain to be addressed.

Primary concerns:

1. Characterization of Vgat-cre expression within the preBötC. A critical reagent used in the manuscript is the Vgat-cre transgene which enables expression of channel-rhodopsin in glycinergic and gabaergic neurons. However, the specificity of expression has not been thoroughly characterized. This is critical because contrary to what may be expected, optogenetic activation of Vgat-cre derived neurons during inspiratory speeds up the breathing rhythm. Two important experiments should be done. First, Vgat-cre;Ai7 labeled neurons should be colocalized with markers for glycinergic and gabaergic neurons. This should be quantified as % of glycinergic neurons that are Vgat-cre positive, % of glycinergic neurons that are Vgat-cre positive, and % of Vgat-cre neurons that are glycinergic and gabaergic. Second, the optogenetic effects observed in Figure 3E should be blocked by the application of gabazine and strychnine to the preparation.

2. Conclusion that activation increased respiratory following activation of inhibitory is not due to post-inhibitory rebound. On page 11 the authors state that the rapid bursting observed following optogenetic activation of Vgat-cre neurons in slice is not due to post-inhibitory rebound. However, the control experiment for this, Figure 3H, is insufficient to make this conclusion. Given that the length of the refractory period is dictated by the extent and length of depolarization, Figure 3H is measuring the probability of post-inhibitory rebound following a normal complete burst. While in comparison, Figure 3E is a prematurely terminated burst after activation of the preBötC Vgat-cre neurons. Therefore, it remains a likely possibility that the refractory period is indeed much shorter after a terminated burst, Figure 3E. The experiment in Figure 3H should be repeated and the refractory period should be measured following a terminated burst as in 3E.

3. Conclusion that Dbx1-derived neurons are distinct from Vglut2 neurons. The manuscript eludes to a population of excitatory neurons that are distinct from the Dbx1-derived excitatory neurons and that these neurons have distinct roles in breathing (Figures 5-7). However, critically, they have labeled Dbx1-derived neurons by injecting tamoxifen into pregnant dams at e10.5 which will induce recombination via the Dbx1-creER transgene. Kottick et al. (2017) Fate mapping neurons and glia derived from Dbx1-expressing progenitors in mouse preBöttinger complex. *Physiol. Rep.* recently reported that

Dbx1-expressing neurons can be labeled by tamoxifen from e8.5 to e11.5. Therefore, by only injecting tamoxifen at e10.5, this work has failed to completely label Dbx1-derived preBötC neurons. Therefore, the conclusion that they are functionally distinct is incorrect since the Vglut2-cre transgene should fully label all Dbx1-derived excitatory neurons. In order to make this conclusion, the authors should repeat their in vivo experiments with the Dbx1-cre transgene to more completely label Dbx1-derived neurons. Furthermore, they should also repeat the in vitro experiments with Vglut2-cre.

4. Conclusion that refractory period of Dbx1 neurons determines burst rate in vitro. Figures 1 and 2 show a nice correlation that the Dbx1 neuron refractory period correlates with the probability of bursting. However, the authors define the refractory period by activating only a subset of Dbx1-derived neurons. This may not be the true refractory period since they have incompletely labeled Dbx1-derived neurons (discussed above). The in vitro refractory period (Figure 2C) should be determined with the Vglut2-cre line. As in Figure 7Ai, it may be possible for bursting to occur earlier in this transgenic line, which would force the authors to reinterpret the endogenous bursting rate and the refractory period.

Secondary concerns:

1. Determining if prematurely terminated bursts in vitro are transmitted through the hypoglossal nerve. In Figure 3E the preBötC activity is measured with an extracellular electrode. This electrode will pick up the ChR2-induced depolarization of Vgat-cre-derived neurons. Therefore, the preBötC burst should be measured by its downstream surrogate, cranial nerve 12 rootlet. It is important to know that the prematurely terminated burst is actually leading to a motor output of a breath.

2. Characterization of Dbx1 depolarization during expiration. In Figure 3F the authors activate Dbx1-derived neurons during expiration. According to their model, if Dbx1 neurons never recover from the refractory period, then a burst should not occur. How is it that Dbx1 neurons recover from the refractory period while they are depolarized by ChR2 during all of expiration. Whole cell recordings of Dbx1 neurons should be performed during the light pulse to characterize the extent of depolarization that occurs during a prolonged light pulse through expiration.

3. Please add quantification for Figure 1F and 1G.

4. Please increase the N for Figure 2A and B. The spread after adding gabazine is quite large and should be more convincing.

5. The text should not state that ZsGreen is GFP. These are different proteins. Perhaps they could call ZsGreen a reporter instead.

We therefore invite you to revise and resubmit your manuscript, taking into account the points raised. Specifically, please address with further control data the specific concerns expressed by referee #4, including but not limited the common concern of referees about the over overlap between Dbx1 and Vglut2 populations. Please highlight all changes in the manuscript text file.

Reviewer #1 (Remarks to the Author):

Baertsch et al study on the “Interdependence of excitation and inhibition in respiratory rhythm generation” deals with the so far puzzling problem of the role played by the large population of inhibitory neurons in the preBötzinger complex. This structure provides the core circuit for respiratory pattern generation. It had been shown earlier that respiratory bursting remains after a blockade of all inhibition, showing that excitatory neurons are sufficient to generate respiratory activity. They combine slice and in vivo work in mice in which ChR had been introduced in either the inhibitory neurons or in the excitatory Dbx- subpopulation or finally in all excitatory neurons in preBötzinger (PB). They show that the inhibitory neurons actually play an important role in regulating the burst frequency and enhancing the frequency range of bursting.

When all inhibition has been blocked, the respiratory bursts increase in amplitude and a slow rate with a long “refractory” phase. Similarly for individual excitatory neurons, if activated by light pulses, they express a long afterhyperpolarisation, which increases markedly with the increase in the preceding light-induced burst of action potentials. If instead the population of inhibitory neurons is activated by pulses of light during a respiratory burst, the burst amplitude will decrease, and the refractory period, following the burst will also shorten and thereby the burst frequency will be increased. When inhibitory neurons are activated during the interval to the next burst, they will as expected instead delay the subsequent burst.

The inference of the burst frequency increase with inhibition during the burst is presumably that the level of activity in the population of excitatory neurons is decreased, and thereby a shorter afterhyperpolarisation (AHP) will result in the entire population of interacting excitatory neurons. Activation of the vagal input (lung stretch receptors) also results in a shorter cycle length presumably through an action on inhibitory neurons. The paper contains a wealth of important information, is well written and the figures to the point.

We thank the reviewer for their positive comments and thoughtful critiques.

Major comments:

The abstract could better convey the results and the authors interpretation. Rather than mentioning “refractory dynamics”, I think the straightforward explanation that the AHP of single excitatory interneurons is markedly dependent on the degree of activation (number and frequency of action potentials) of single excitatory interneurons (Fig 1H, I), and the population as well. Regulating the degree of PB excitatory interneuron activity by the inhibition will automatically regulate the duration of the AHP in the population of cells.

We agree with the reviewer and have revised the abstract for improved clarity.

One problem, perhaps handled at a downstream level, is that whereas the level of PB activity decreases with increased burst frequency, the net output to the respiratory muscles during inspiration increases with respiratory rate. Please comment!

This is an excellent point that remains an important open question. Although high frequencies are often associated with smaller tidal volumes (e.g. sniffing), and low frequencies with larger tidal volumes (i.e. sighing, gasping); it is certainly possible to increase frequency and tidal volume simultaneously, for example during hypercapnia. There are many mechanisms that could allow increased motor output to occur despite decreased preBötC activity, including, as the reviewer points out, downstream mechanisms at the level of premotor and/or motor neuron pools, which can be modulated to change gain- producing more output for a given input. We have added the following statement into the discussion: “Although mechanisms downstream of the rhythm generating circuit can further shape respiratory motor outputs, we find that hyperexcitability...” This is an interesting topic that we would like to pursue in the near future.

The results in Fig 2C (page 8 and also 10 (Fig 3)) showing the effect of light pulse activation of Dbx neurons

during the refractory period are to the point, but it would be more informative to plot them as phase-response curves (plotted in relation to an unperturbed cycle). In this context it would also be important to report the exact timing of the inhibitory light pulse within different parts of the PB-burst. Whether the effect will differ in the initial, middle or late part of the PB-burst – could be clearly represented in a phase response curve. As well as the effects during the refractory period.

We thank the reviewer for their comment, but we feel that the all or nothing nature of evoked bursts in vitro is better represented as a probability. Therefore, we wish to keep the figures demonstrating the refractory period as is. However, to address the reviewers comment, we now include phase shift plots for the in vitro experiments in Supplementary Fig. 1.

The legends are in general too brief – often lack of sufficient information to critically evaluate the figures – for instance legend Fig 4. In Fig 4 E, one has to guess which bar is what, and the meaning of 4H is not obvious. What is CPAP? The manuscript and figures should be understandable for a general neuroscience readership particularly in this journal

We agree with the reviewer, and have revised the legends to include as much information as possible within the word limits. In some cases, the figures have also been revised for clarity.

Page 12-13 Effect of vagal input.

The authors should refer to the cat data of Clark and Euler 1972 showing that vagotomy causes a marked slowing of the respiratory burst rate to a constant slow rate rate. Varying the CO₂ drive changed the amplitude at the output level (phrenic nerve) but did not affect the burst frequency. Also consider this in relation to the current results.

We thank the reviewer for pointing out this oversight. These studies are now referenced in the revised manuscript.

Minor comments:

Page 9 lines 177-184

Figure S2A shows a neuron and population recordings for what is referred to as a fictive sigh, which has a somewhat longer burst interval and a larger afterhyperpolarisation. It would seem important to define the criteria better for what is considered as a fictive sigh, rather than what seems just an unusually large burst followed by a longer refractory phase. If this cannot be done – this part could be omitted.

We thank the reviewer for noticing that we did not sufficiently define the sigh in this particular study. Criterion distinguishing a fictive sigh from larger amplitude bursts have been well described previously by our group and others. Most notably, sighs are biphasic, with a large burst superimposed on an initially normal “eupneic” burst. This is now described in more detail in the text with additional citations and described in the legend for Supplementary Fig. 3.

We feel considering refractory dynamics in the context of the sigh lends strong support to our overarching hypothesis and should be included in the study, particularly given recent interest in the sigh (Li et al., 2016).

Page 13, lines 270 to 280,

This would seem to fit better in Methods.

To address the reviewer’s comment, we have shortened this section by removing “The basilar artery, caudal cerebellar artery, and intersection of the vertebral arteries were used as landmarks (Figure 4g). To account for small variations in the size of animals, distances were determined relative to the half width of the brainstem at the level of the caudal cerebellar artery.” However, we feel it is important to show the location and consistency of our stimulation sites in a display figure. As such, these data must be briefly referred to in the results section.

Page 20 line 431

Why “limiting synchronization” rather than just increased overall activity in the PB population – due to excitatory interaction among the excitatory PB neurons. The higher the level of activity in each neuron – the larger AHP and refractory time.

We agree with the reviewer and have substantially revised the discussion to improve clarity and to accommodate the word limits of Nat Commun. Mentioning synchronization is important since it has been

shown the inhibitory neurons reduce synchrony in the network, which will be a major contributor to their overall effect of reducing network activity and the refractory period.

Page 20, lines 437-439.

Should Cheyne-Stokes breathing be relevant in this context?

The reviewer raises a very interesting question! Cheyne-Stokes breathing is generally thought to arise from disturbances in the gain of chemosensory feedback leading to waxing and waning of breathing. However, it is very possible that changes in the balance of excitation and inhibition at the level of the preBötC could be a mechanism contributing to increased chemosensory gain. However, due to space limitations, and a central contribution is somewhat speculative, and because we did not experimentally study Cheyne-Stokes breathing, we feel it is best not to explicitly address this issue in the discussion.

Page 21 lines 452 to 461

Sr has an additional effect in that it blocks the Ca²⁺ site internally on Kca channels, which could explain the findings.

The reviewer makes an excellent point! If KCa channels contribute to afterhyperpolarization, Sr should reduce the refractory period and increase frequency by blocking the afterhyperpolarization. However, since there are many other potential cellular mechanisms that could contribute, we have removed these nuanced details in the revised Discussion. We are currently investigating the underlying cellular mechanism(s) giving rise to the refractory period, but it is too early to provide any details, and address this important issue in this manuscript.

Reviewer #2 (Remarks to the Author):

I read this manuscript three times to prepare this review because it contains a great deal of data and covers a lot of ground. The manuscript addresses important contemporary problems that pertain to central pattern generation as well as sensorimotor integration. It could make important contributions to the field(s) of CPGs and respiratory neurobiology. However, I think there are a few interpretive problems that must be dealt with as well as many other items that require clarification.

Addressing the role of GABAergic synaptic inhibition in respiratory rhythm, the authors demonstrate that fast inhibition can operate on a cycle-to-cycle basis to modulate the amplitude and frequency of respiratory rhythm. Modulation of respiratory (breathing) frequency is the main focus; the authors compare to what extent inhibitory vs. excitatory interneurons can influence frequency. They find that inhibitory mechanisms are more powerful, which is a new insight of general interest.

The effect of fast synaptic inhibition on the amplitude of breaths has been demonstrated before for glycinergic neurons (Sherman et al. Nat Neurosci 18: 408-414, 2015) but the authors here address GABAergic neurons. It is possible that these are the same or overlapping populations that co-express GABA and glycine. I recommend that the authors comment on that in Discussion.

As for frequency, Sherman showed that activation of glycinergic preBötC neurons during inspiration attenuates the inspiratory breath and phase advances the core oscillator. Those authors applied single brief pulses to Glyt2 mice injected with AAV to express ChR in the preBötC and constructed the phase-response curve. The present manuscript uses Vgat2-ChR2 transgenic mice and applies a more sophisticated approach by triggering light pulses off of the inspiratory waveform for repetitive cycle-triggered photo-stimulation. Doing so, they show that phasic inhibition very potently modulates frequency and enables the 2-3 Hz breathing frequencies characteristic in vivo. Interestingly, just photo-stimulating excitatory neurons is not enough; the frequency only increases a little bit. So, the results are an advance compared to Sherman, although not totally novel. Again, commenting on the potential overlap between GABA and glycine preBötC neurons would be helpful so the reader can interpret the present study and the Sherman study in context (are they the same or overlapping populations, or separate populations that accomplish similar tasks? Are GABA and glycine preBötC interneurons redundant?)

We agree with the reviewer, that our study builds and extends the observations made in the excellent study by Sherman et al. But, the phase-specific stimulation and the stimulation of glycinergic and GABAergic neurons allowed us to reach novel insights. We now comment on this in the discussion.

However, we would like to point out, that detailed comparisons with Sherman et al with regard to the effects of stimulation of glycinergic vs. glycinergic+GABAergic neurons are difficult due to differences in the preparations used. This will be further discussed below.

Another part of the manuscript looks at intrinsic refractory mechanisms in the preBötC. The PI (Ramirez) has carved out an interesting problem, that of post-inspiratory mechanisms. Recently his group identified the “post-inspiratory complex” (PiCo). This paper follows that line of inquiry. They show that synaptic inhibition influences the preBötC refractory period and thus breathing frequency; when inhibition is blocked, the refractory period is enhanced, frequency is correspondingly diminished.

The phase-response curves in Fig. 7Bi and Bii are the best part of the paper (it's a shame they come so late). The steep purple curve in the vagus intact mouse shows how sensitive the phase (frequency) is to inhibition. That curve gets quite flat during the refractory period in the vagotomized mouse. This is evidence that supports the claim that sensory feedback from lung mechanoreceptors, i.e., one major source of inhibition, does influence burst termination. When that feedback inhibition is removed (by vagotomy) the preBötC neurons generate larger bursts, making them refractory either by synaptic depression or intrinsic activity dependent outward currents, and thus become unresponsive during that refractory period, hence the flat phase-response curve after vagotomy (in response to Vgat2-ChR2 photo-stimulation).

Well said! We now point this out in the Results. As for the order of the figures, we thought about this, but don't see an easy possibility to show the response curves earlier without disturbing the flow of the entire text.

My enthusiasm for the paper notwithstanding, I do have several problems that should be addressed:

We thank the reviewer for their enthusiasm and appreciate the time spent to provide a detailed assessment of our manuscript.

The block of inhibition in previous slice studies did NOT modulate XII/preBötC burst frequency. That result has been widely replicated. A particularly comprehensive report by Gray et al. Science (1999) shows that a variety of peptides modulate respiratory frequency, but that fast chloride-mediated inhibition does not. And, they showed that peptide modulation of the rhythm stays the same whether fast inhibition is present or not (see Fig 1B in Gray et al Science 286:1566, 1999). There are many other studies from the Smith lab, Ballanyi (Brockhaus & Ballanyi), and Greer (Ren & Greer). Why are these present results so different from past studies?

It is true that previous studies have reported mixed effects on frequency by pharmacological removal of synaptic inhibition; some with small increases (e.g. Shao and Feldman, 1997), decreases (e.g. St-John et al., 2009) or no change (e.g. Gray et al., 1999; Brockhaus and Ballanyi, 1998). And, the frequency effect of inhibition block can vary in vitro based on extracellular [K] (Del Negro et al., 2009) and postnatal age (Ritter and Zhang, 2000).

As we point out in our manuscript, the preBötC contains populations of inhibitory neurons with distinct phasic activities that have opposing influences on respiratory frequency: Reduced activity of inspiratory inhibitory neurons is predicted to slow frequency by increasing the refractory period, whereas reduced activity of expiratory inhibitory neurons will increase frequency by allowing excitatory neurons to synchronize sooner. Thus, the net-effect on frequency during non-specific blockade of inhibitory mechanisms in brainstem slices will depend on the relative influence of inhibitory neurons with inspiratory vs. expiratory phasic activities. Since a larger proportion of inhibitory neurons with expiratory activity are thought to be located more rostrally in the BötC, variations in the exact rostral to caudal location of brainstem slices is likely to shift the net effect of inhibition and may help explain these discrepancies.

Most importantly, however, many studies characterized the respiratory output recorded in the XII. As the reviewer will agree, this does not necessarily reflect the activity in the preBötC. There is ample evidence that the preBötC activity does not always transmit to the XII motor output in slices (just to name two examples: Ramirez et al., 1996; Kam et al., 2013). Thus, the results achieved by blocking synaptic inhibition, will vary depending on whether the frequency was quantified at the level of the preBötC or the XII. Moreover, as shown here and in other studies blocking inhibition increases activity of excitatory preBötC neurons, slowing the rhythm observed in the preBötC, but also improving transmission of preBötC bursts to XII. Thus, increased transmission to the XII could lead to the conclusion that respiratory frequency measured from the XII is increased, despite reduced frequency at

the level of the rhythm generator. Space limitations set by the Journal do not allow us to fully expand on this issue and discuss in the detail all the different publications in which inhibition was blocked in the respiratory network. However, we now state in the discussion “Most previous studies have inferred frequency changes during blockade of inhibition by measuring XII motor output. Note, however, transmission to XII depends on the excitability of the preBötC which will be altered by blockade of inhibition and may therefore skew the interpretation of frequency changes.”

With regard to the peptidergic control of frequency referred to by Gray et al., 1999, it is possible that the action of these peptides modulates refractory period mechanisms through second messenger systems. Indeed, this is an exceedingly interesting possibility (not only with regards to peptidergic, but also aminergic modulation) that we hope to address in the future. Lastly, there could be differences in the potency/specificity of Bicuculline/Picrotoxin used in the majority of previous studies vs. gabazine used in ours. Bicuculline for example, is known to block potassium currents. This could cause additional excitatory network effects, thus complicating the interpretation.

The authors use two transgenic Cre- driver mice for in vivo studies. One is linked to Dbx1 and the other is linked to Vglut2. In the preBötC these Cre- drivers should target largely the same population of neurons, but outside the preBötC there will be many more Vglut2 excitatory neurons than just those derived from Dbx1-expressing precursors. Furthermore, outside preBötC, Dbx1-derived neurons can be excitatory or inhibitory. Therefore, in the optogenetic protocols in Figure 7, where photo-stimulation of Vglut2-ChR2 evokes bursts in the refractory period of Dbx1 neurons, the effect is quite credibly explained by a plethora of ChR2-expressing Vglut2 synaptic terminals in the preBötC from cell bodies located elsewhere. There is NO reason why one would suspect a latent population of non-Dbx1 excitatory neurons in the preBötC. The results could be explained simply by activating a lot of Vglut2 terminals, which is sufficient to overcome the refractory state and evoke a burst.

Acknowledging this alternative explanation is absolutely critical, and then adjusting the interpretation and conclusion accordingly.

We agree with the reviewer, this is an important issue. Bouvier et al., 2010 demonstrated that ~50% of cells expressing Vglut2 co-label for Bgal in Dbx1LacZ/+ mice (all Dbx1 derived neurons expected to be labelled) (Figure 3U), whereas 82% of Bgal neurons co-label with Vglut2. Thus, although the majority of the Dbx1 neurons within the preBötC are glutamatergic, not all glutamatergic preBötC neurons are Dbx1 derived. Indeed, these authors explicitly state “About half of the Vglut2 expression in the preBötC area was accounted for by Dbx1-derived cells...”. This is consistent with the observations of Gray et al, 2010 (See figure 2A) and Gray, 2013 (See figure 9B) in which not all glutamatergic neurons in preBötC co-label with markers for Dbx1-derived cells. Indeed, Gray et al., 2010 explicitly state: “Dbx1-derived neurons, however, only account for a subset of VLM glutamatergic neurons”. Indeed, glutamatergic neurons from other lineages, such as Lmx1B are also contained within the borders of the preBötC (Gray et al., 2010). Therefore, we respectfully disagree with the reviewers’ suggestion that there are not non-Dbx1 glutamatergic neurons in the preBötC.

To address the reviewers’ second concern, we now include the alternative explanation that activation of Vglut2 terminals innervating the preBötC during Vglut2-ChR2 stimulation could increase the activation of rhythmogenic neurons compared to Dbx1-ChR2 stimulation. Specifically, we now state in the Discussion “It is possible that Vglut2-ChR2 stimulation leads to greater excitation of the same subpopulation of neurons due to activation of Vglut2+ terminals innervating the preBötC. Although this may be a possibility, it is difficult to reconcile how the differential effect could be limited to the vagus intact condition, to changes in burst frequency vs. amplitude (Fig.7c), and to the expiratory vs. inspiratory phase (Fig. 8c).”

A lesser concern: the authors should consider that photo-stimulation in Vglut2-ChR2 mice will activate only neurons whereas in Dbx1-ChR2 mice the photo-stimulation protocol will also activate Dbx1-derived glia, which could be 30-40% of the cells undergoing Cre-mediated recombination. There could be purinergic effects related to gliotransmission and perhaps some other influences not yet widely understood.

Yes, we agree with the reviewer! We agree that photostimulation in Dbx1-Cre animals will activate ChR2 expressed by some glial cells and we now mention this in the revised text. Thus, we also combined this approach with the use of the Vglut2^{cre} line, which will not target glia.

Data describing the role of preBötC glia are somewhat limited. However, it seems that increased glial activity (including purinergic signaling) increases frequency. Thus, if anything, our conclusion that excitatory Dbx1 mechanisms alone are not sufficient to generate rapid frequencies may be even more dramatic if isolated to Dbx1 neurons.

The authors frame their conclusions in terms of excitatory and inhibitory mechanisms that control rhythmogenesis. I recommend they distinguish rhythm generation (due to excitatory synaptic mechanisms) from its regulation (due to inhibitory synaptic mechanisms). The two are more easily understood, in my view, once thus distinguished.

We respect the reviewers view. However, in our opinion, excitatory and inhibitory mechanisms are intimately intertwined and, in the functional network, never operate independently, as shown by the concurrent activation of inhibitory and excitatory neurons during each inspiratory cycle. Thus, this may boil down to what is considered respiratory rhythm? Artificially stripping the network of inhibitory interactions reveals very slow rhythm which is non-physiological. Can this be considered representative of respiratory rhythm generation? Is this any more relevant to respiratory rhythm generation than the activity of a single artificially isolated endogenous bursting neuron that continues to produce a rhythm in the absence of other interactions? Although synaptic excitation initiates bursts and therefore generates the respiratory rhythm, we view the breathing rhythm as an emergent “interdependent” property of interacting excitatory and inhibitory neurons that endows the network with the dynamic attributes that characterize breathing (the major thrust of the paper). As such, framing our conclusions in terms of rhythm generation vs. regulation could, in our view, be misleading for the general reader.

That being said, we have softened emphasis on “rhythm generation” throughout the revised manuscript.

SPECIFIC MAJOR CONCERNS

Line 34: Excitation and inhibition are not interdependent for respiratory rhythm generation. Synaptic excitation is obligatory for rhythmogenesis, whereas synaptic inhibition is dispensable. The line in the Abstract as written is misleading. Inhibition is relevant for modulating the timing and the magnitude of inspiratory bursts, which influences the rhythm but that does not mean that they are “interdependent for respiratory rhythm generation”.

We agree with the reviewer that inhibition is not obligatory for the respiratory rhythm to persist, something that has been demonstrated numerous times (including by our laboratory). Indeed, we demonstrate this once again in the present paper in vitro and in vivo (Figs. 3 and 6). However, we feel it is misleading for the general reader to emphasize these mechanisms as obligatory or dispensable as it paints the picture that dispensable mechanisms are less important. To the contrary, a major implication of our study is that inhibition is critical within the functional network to generate respiratory rhythms relevant for physiological breathing. In this sense, we view excitation and inhibition as “interdependent” for the breathing rhythm. However, we have revised the manuscript to soften the focus on respiratory rhythm generation per se. E.g. we have changed the title to “The interdependence of excitation and inhibition for the control of dynamic breathing rhythms”.

Lines 128-131: the cell in Fig. 1G hyperpolarizes a lot (>5 mV) during post-inspiration, which is a characteristic presciently identified by Reklung et al. J Neurophysiol 75: 795-810, 1996. However, Reklung et al. pointed out that these so-called Type 1 preBötC neurons were approximately half of the inspiratory neurons they recorded. Picardo et al. (2013) working specifically on Dbx1 preBötC neurons obtained largely the same data and came to a similar conclusion that the preBötC neurons with large post-burst after-hyperpolarizations (AHPs), the so-called Type 1 neurons, were about half of the Dbx1 preBötC population. My question: Do the authors find ~50% of their recorded Dbx1 neurons like Fig. 1G with the deep post-inspiratory AHPs or is their fraction higher (or lower). And, should that be acknowledged in Results?

We thank the reviewer for this great suggestion. We now include a quantification of the AHP in Dbx1 neurons (Fig. 2c). Although we found considerable variability in the magnitude of the AHP, we did not observe a clear distinction between two “types” of neurons with respect to the AHP. Instead, we found that the AHP was related to the depolarization (drive potential area) of the neuron during inspiratory bursts. This information is now included in the Results and discussed in the text.

Lines 208-209: The optogenetic pacing protocol here is a clever demonstration that inhibition can influence frequency. Nevertheless, the experiment itself shows the same qualitative point as Sherman et al. Nat Neurosci 18(3): 408, 2015, who showed that optogenetic activation of glycinergic (Glyt2, Slc6a5) preBötC neurons phase-advances the next breath. Here, by repeatedly applying photo-stimulation to Vgat+ preBötC neurons the authors just show that photo-stimulation phase advance works repeatedly and the rhythm speeds up as a result. This point largely recapitulates a point I made above in general comments.

We agree with the reviewer, the study by Sherman et al., Nat Neurosci examined the phase-dependent effect of stimulating preBötC glycinergic neurons. However, the Sherman study provided no mechanistic explanation for why inhibition during inspiration advances the next breath. Moreover, this study did not systematically compare the ability of inhibitory vs. excitatory mechanisms to control respiratory frequency. Thus, we feel the importance of the balance of excitation and inhibition in the context of refractory mechanisms is a significant conceptual advance.

Lines 214-218: Why don't the authors show these data regarding graded activation either in a main figure or supplemental?? I strongly encourage them to show those data.

As suggested by the reviewer, we now include these data in Supplementary Fig. 6.

Lines 218- 232 and Fig 3: I recommend that authors annotate the charts in Fig. 3G so they are a little easier to decipher on sight. The data are important but it is hard to know what all the measures and categories are just from the figure itself. Consider how the graphics can be improved so the reader does not have to stop and consult the legend.

We agree with the reviewer and have revised this figure to try make it easier to interpret on sight. For example, we now include insets to easily distinguish unilateral vs. bilateral stimulation, and rearranged the graphs to align inspiratory and expiratory stimulations of Dbx1 and Vgat groups.

Lines 237-241: Did the authors ever patch-record a Vgat- neuron (in the Vgat-ChR2 mouse) while trying to evoke a preBötC burst by triggering post-inhibitory in the Vgat+ population? It would bolster the point they are making if they could show that the post-inhibitory rebound does NOT occur in the ~2 s refractory period. The field recording just shows that the preBötC burst does not occur during the ~2 s refractory period.

Yes, we agree with the reviewer! We now include an example recording of a Vgat- neuron during stimulation of inhibitory neurons where rebound spiking is reduced closely following a burst (Fig. 5d). This is also apparent in the inset of Fig 5b. Notably, the membrane potential still exhibits rebound, yet the neurons do not sufficiently spike to evoke a burst at the population level. This is reminiscent of results described during current injection during the refractory period (Fig 2e).

Lines 277-278: The authors write, "Extracellular recordings in this area confirmed inspiratory population activity that occurred prior to XII bursts". Can they show a trace to demonstrate that?

As suggested by the reviewer we now included an example trace in Fig. 6f.

Lines 350-351: This is over interpretation. These mice are Vgut2-Cre;ChR2 mice, therefore all the excitatory axons of passage that enter in and synapse within the preBötC are evoked by light stimulation. The phase advanced inspiration in Vglut2-ChR2 mice (which does not occur in Dbx1-ChR2 mice) could be due to exciting so many excitatory ChR2-expressing synaptic terminals in the preBötC which then excite the core rhythm generators and overcome whatever refractory mechanisms are ordinarily preventing the light evoked burst in the ~200 ms refractory period in vivo. This largely recapitulates a general critique above.

To address the reviewer's comments, we have changed this section substantially in the revised MS, and we now include the alternative explanation that activation of Vglut2 terminals innervating the preBötC during Vglut2-ChR2 stimulation could increase the activation of rhythmogenic neurons compared to Dbx1-ChR2 stimulation. However, this interpretation does not account for the finding that the differential effect of Dbx1 and Vglut2 stimulation are isolated to the vagus intact condition, to expiratory vs inspiratory phase, and to frequency vs amplitude. We more clearly state these considerations in the Results and Discussion.

Specifically, we state “Although stimulation of Dbx1+ and Vglut2+ populations had indistinguishable effects in vitro (Fig. 2b, Supplementary Fig.1a) and in vagotomized mice in vivo (Fig. 7b, 8b, 9b), with vagal sensory feedback intact, the refractory period for Vglut2-ChR2 was reduced relative to Dbx1-ChR2 (Fig. 9a). It is possible that Vglut2-ChR2 stimulation leads to greater excitation of the same subpopulation of neurons due to activation of Vglut2+ terminals innervating the preBötC. Although this may be a possibility, it is difficult to reconcile how the differential effect could be limited to the vagus intact condition, to changes in burst frequency vs. amplitude (Fig.7c), and to the expiratory vs. inspiratory phase (Fig. 8c).”

Lines 462-466: I'm confused by the statement “We identified an alternative refractory mechanism by recording from synaptically isolated preBötC neurons; i.e. a membrane property that leads to hyperpolarization and reduced spiking following network bursts...” That refractory mechanism was first described by Reikling et al. J Neurophysiol 75: 795-810, 1996. It is not novel or alternative because it was earlier proposed to influence the respiratory rhythm generation mechanism.

It was not our intention to imply that we discovered the afterhyperpolarization of these neurons, which indeed has been shown by multiple authors, and we have revised the language throughout the MS accordingly. However, to our knowledge, the observation and quantitative analysis that this afterhyperpolarization limits subsequent spiking in an amount proportional to burst activity has not been described. Also, in our opinion, interpreting the afterhyperpolarization in the context of a mechanism that limits the ability of Dbx1 neurons to generate bursts during the refractory period is also conceptually new.

Lines 473-474: Regarding the conclusion, “one would expect a period of reduced spiking during the refractory period to decrease burst frequency.” Not necessarily. That presumes that recovery from refractoriness and burst initiation are linked. That may not be the case as synaptic vesicles restore and reset, and baseline membrane comes back to its baseline (cellular autonomous mechanisms), and then a process of percolation of excitatory synaptic activity initiates the next burst (network properties attributable to topology and number of interconnections among the constituent rhythm generators).

We agree with the reviewer, that the interplay between refractoriness and percolation within the network is an interesting and potentially complex issue. However, the recovery from refractoriness and burst initiation are likely linked since, by definition, the refractory period is the time during which the rhythm generating neurons cannot initiate a burst. Indeed, our data suggest that refractory mechanisms occur in the same neurons that initiate bursting and therefore percolation among excitatory neurons may not effectively occur during the refractory period because spiking in these neurons is largely suppressed. Thus, “one would expect a period of reduced spiking during the refractory period to decrease burst frequency” precisely because it delays the onset of percolation.

Although percolation can be slowed (e.g. by inhibitory neurons with expiratory activity) or accelerated (e.g. via increased excitation), our results suggest that the duration of T_e and thus the control of frequency is dominated by the refractory period (particularly in vivo), and that percolation occurs relatively rapidly. For example, T_e is dramatically shortened following activation of inhibitory neurons during inspiration, which is not expected to alter the subsequent percolation per se, but will reduce the refractory period.

For these reasons, we don't believe our statement conflicts with the reviewers view, however, we have modified the text for clarity in the revised manuscript. We now more specifically state” Moreover, in a network thought to rely on recurrent excitation for burst initiation^{72,73}, reduced excitability during the refractory period likely delays the onset of excitatory interactions percolating through the network, thereby increasing the period between bursts.”

Lines 499-500: The authors conclude that stimulating Vglut2+ neurons evokes bursts during the Dbx1+ neurons' refractory period. However, it could be Vglut2+ synaptic terminals in the preBötC, whose cell bodies could be located anywhere, which evoke the burst. The vast majority of Vglut2 neurons in preBötC are Dbx1-derived, but there could be many more Vglut2 inputs to preBötC not Dbx1-derived. Again, this recaps a point in general comments.

Please see above.

Lines 502-503: In conjunction with the comment above, it is not possible on the basis of these data to conclude that non-Dbx1 excitatory interneurons can be rhythmogenic when the photo-stimulation effects, i.e., the ability to evoke bursts during the refractory period could be attributed to Vglut2+ terminals in the preBötC and NOT neurons located therein. This conclusion is not credible.

The discussion has undergone substantial revision and we have removed this statement from the text to address the reviewers concern.

SPECIFIC MINOR CONCERNS

Lines 54-57: Certainly the preBötC serves as the “master clock” as the authors indicate, but its preeminent role is inspiratory rhythm generation. For the non-specialist, the “master clock” being introduced first might be misleading.

We agree with the reviewer, and have revised this statement accordingly. We now state: “An ideal model system for studying rhythmicity is the preBötzinger Complex (preBötC), an autonomous oscillator located bilaterally in the ventrolateral medulla that generates the inspiratory phase of breathing (Smith et al., 1991).”

Lines 83-84: I recommend the authors also cite Kuwana et al. Eur J Neurosci 23: 667-74, 2006 because it characterizes GABAergic preBötC neurons, whereas Winter et al. (2009) deals with glycinergic neurons.

We thank the reviewer for this suggestion! This is an important paper that we now cite throughout the revised manuscript.

Lines 106-107: Tamoxifen-inducible Cre expression in Dbx1-CreERT2 mice, and its effect on fusion protein expression in the preBötC of intersectional offspring has very recently been quantified in detail by Kottick et al. Physiol Rep 5(11): e13300, 2017. I recommend the authors consider citing it.

We thank the reviewer for this suggestion. Kottick et al., 2017 has now been referenced multiple times in the text.

Line 142: “The magnitude of reduced spiking...” is confusing. Consider rephrasing, e.g., “diminution of spiking ability”?

We agree with the reviewer, and have substantially revised this section of the manuscript. We now state “The reduction in spiking activity was related to the intensity (Fig. 2e) and duration (Supplementary Fig. 2b) of the simulated drive potential. Synaptically isolated neurons were identified as Dbx1+ with depolarizing responses to light. Notably, spiking was also reduced following current injection in isolated Dbx1- neurons (Supplementary Fig. 2c), suggesting refractory mechanisms may be similar for Dbx1+ and Dbx1- preBötC neurons.”

Lines 177-178: Why aren't the authors referring to some important in vitro ‘sigh’ work by their group Lieske et al. Nat Neurosci 3: 600-7, 2000, and others, Ruangkittisakul et al. J Neurosci 28: 2447-58, 2008?

As suggested by the reviewer, we now reference these important studies regarding reconfiguration of the preBötC to generate sighs.

Line 193: The 3D effect on the pie chart is superfluous ink (see: Tufte, ER. Visual display of quantitative information. Graphic Press: Cheshire, CT, 2001); i.e., the 3D effect does not enhance scientific communication. Also, the graphic itself is superfluous: having the fractions written in the text is enough, the pie chart does not make the point more clear in my view.

The 3D effect on the pie charts has been removed. However, we would like the quantified data on these inhibitory populations displayed so that the rest of the figure can be easily interpreted without having to refer to the text. This figure has been revised.

Lines 196-197: It is interesting that expiratory neurons became inspiratory (Fig. 3C here). Consider Shao & Feldman J Neurophysiol 77: 1853-60, 1997, who showed in slices that BIC or STR applied separately abolished expiratory activity in preBötC neurons but did not induce inspiratory modulation per se. On the other hand, zero-chloride ACSF made the XII nerve go tonic but seems to have made the expiratory neuron

discharge what “seem” like bursts. Similarly, Brockhaus & Ballanyi Eur J Neurosci 10:3823-39, 1998, showed the expiratory-to-inspiratory transformation solely in response to BIC or STR. These precedents in the literature should be dealt with and might bolster their present result.

We thank the reviewer for this good suggestion and now cite the suggested studies. However, because of the word limits of the journal, we have not included detailed discussion about these results in the text.

Previous studies suggest that preBötC neurons receive different ratios of concurrent excitatory and inhibitory inputs during inspiratory bursts, and this likely determines the phasic activity and burst shape of each neuron. When inhibition is stronger than excitation, the neuron becomes expiratory, and blockade of inhibition can reveal weak or strong inspiratory activity depending on the amount of excitatory input. Thus, we expect results from individual expiratory neurons will be quite variable.

Lines 267-269: It's already well established that vagal afferents act on the preBötC via inhibitory mechanisms.

Yes, we agree with the reviewer. This has been done previously, but is important to replicate since it is critical for interpreting how the loss of vagal feedback changes activity in the preBötC and the refractory period. We have revised the language to better reflect that these results confirm prior observations.

Line 273: recommend write “half”, instead of the numeral fraction.

As suggested by the reviewer, this section has been revised and ½ has been changed to “half” throughout the manuscript and figures.

Lines 298-299: What are the non-specific effects here? I don't clearly see what they might be from Fig. S6.

At high laser powers, light stimulation in control mice lacking Cre increased XII amplitude and decreased frequency in some animals. These non-specific effects are minimal in the average data shown in Figure S6, but nonetheless became obvious at powers <7mW.

This information has been moved to the methods section to comply with the word limits of Nature Communications.

Lines 374-375: This is an interesting observation (Fig. 7Bii) that activation of inhibitory neurons has little effect during the refractory period. That seems to be more evidence that the refractory period is attributable to synaptic depression or activity dependent outward currents in rhythmogenic preBötC interneurons.

We agree with the reviewer. This point has been further emphasized in the revised manuscript. Specifically, we state “...in vagotomized mice, light pulses had little effect during the time following inspiration that corresponded to the refractory period (Note the flat area of the curve in Fig. 9d), suggesting activity of inhibitory neurons during the refractory period has minimal effects on breathing (also see Fig. 5d).”

Line 494: Regarding Dbx1 pattern-generating neurons in the preBötC, Cui et al. Neuron 2016 is an important citation, but I recommend also citing Wang et al. eLife, 2014, who demonstrated Dbx1 premotor neurons in the preBötC.

This study is now cited as suggested.

Line 561: The Dbx1 Cre-driver mouse line is described as tamoxifen inducible, with the “ERT2” appended to its nomenclature, in the Methods and earlier in the manuscript. The acknowledgement lists it simply as Dbx1-Cre. I presume the mouse line is constant so I recommend the authors append ERT2 in the acknowledgement.

We thank the reviewer for this recommendation. This has been corrected as suggested.

Reviewer #3 (Remarks to the Author):

The work of Feldman and colleagues supports the notion that the generation of the inspiratory oscillations in the preBoetzinger complex (preBötC) results from an all excitatory circuit. The latest incarnation of this work suggest that each excitatory cell tends to produce a subthreshold burst (burstlet) , and the next cycle of

oscillatory output occurs when a critical number of burstlets occur at the same moment. This is reminiscent of oscillators in the inferior olive (IO).

Here, Ramirez show that inhibition is critical in setting the frequency of the oscillations, again reminiscent of what occurs in the IO. This result are extremely important as there were no prior data to suggest how the preBotzinger can oscillate over a very wide range of frequencies - easily 3 to 10 Hz in the rat. Further, the authors showed that oscillations at the sniffing frequency, synonymous with coordinated orofacial behavior, are completely dependent on inhibitory input.

Further, Ramirez and colleagues show that the mechanism is a control of excitation to reduce the refractory period, presumably by limiting the number of cells that are inactivated and drop out of producing burstlets. This is a very interesting control mechanism and one that has not been previously discussed in the CPG literature, which is dominated by data on oscillators with a fixed frequency.

Lastly, the authors assess the influence of the vagia nerve.

Suggestions to improve the manuscript.

1. "Rhythmicity is generated by neural networks through a combination of precisely timed excitatory and inhibitory mechanisms ..." I think this should be revised since the mechanism for rhythmicity in spike generation in the PreBötC and the IO is just excitation.

We agree with the reviewer that this will likely be a general mechanism. To reflect the general nature of this phenomenon, we have revised this section to now state: "Rhythmicity often emerges from neural networks containing a combination of excitatory and inhibitory neurons."

2. The data of Figures 1F and 2B implies that the authors can construct a phase resetting curve for (transhemispheric synaptic) excitation, i.e., the shift in period to the next burst relative to when the perturbation occurs relative to the unperturbed period (as estimated from previous cycles); like figure 7Bi, but with phase as the abscissa. This addition, presumably calculated from existing data, would be a valuable contribution for anyone hoping to model the system and possibly map it into a network of coupled (noisy) single-cell oscillators.

We thank the reviewer for making this good point. As suggested by the reviewer, we now include phase shift plots for our in vitro data in Supplemental Fig. 1a,b.

3. Similar to the above, the data of Figures 3H implies that the authors can construct a phase resetting curve for synaptic driven inhibition. They should do so.

See above.

In toto, the topic is important, the experiments are far from trivial, the data appear clean, and the logic of the study seems solid. I vote for publication.

We thank the reviewer for their kind compliments and thoughtful assessment of our work!

Reviewer #4 (Remarks to the Author):

The manuscript titled "The interdependence of excitation and inhibition in respiratory rhythm generation" by Baertsch et al. is a series of well-designed and thoughtful experiments to determine the general function of inhibitory neurons in the breathing rhythm generator, the preBötC. The preBötC is composed of several thousand cells and half of these are inhibitory. However, because inhibitory neurons are not required for rhythm generation, their role in breathing has remained largely uncharacterized. Baertsch et al. use general transgenic lines to determine a role of inhibitory neurons in limiting the extent of depolarization of excitatory inspiratory neurons. The model predicts that the frequency of breathing is dictated by the refractory period of the preBötC following a burst and that inhibitory neurons can speed up the frequency of bursting by limiting the extent of excitation, thereby reducing the refractory period. Although the data and combination of in vitro and in vivo experiments are quite compelling, 4 primary concerns and 5 secondary concerns remain to be addressed.

We would like to thank the reviewer for their supportive remarks and in-depth review.

Primary concerns:

1. Characterization of Vgat-cre expression within the preBötC. A critical reagent used in the manuscript is the Vgat-cre trasngene which enables expression of channel-rhodopsin in glycinergic and gabaergic neurons. However, the specificity of expression has not been thoroughly characterized. This is critical because contrary

to what may be expected, optogenetic activation of Vgat-cre derived neurons during inspiratory speeds up the breathing rhythm. Two important experiments should be done. First, Vgat-cre;Ai7 labeled neurons should be colocalized with markers for glycinergic and gabaergic neurons. This should be quantified as % of glycinergic neurons that are Vgat-cre positive, % of glycinergic neurons that are Vgat-cre positive, and % of Vgat-cre neurons that are glycinergic and gabaergic. Second, the optogenetic effects observed in Figure 3E should be blocked by the application of gabazine and strychnine to the preparation.

As suggested by the reviewer, we now include additional control data in Supplementary Fig. 6b and 8d demonstrating that the effects of Vgat-ChR2 stimulation are lost following application of specific blockers of inhibitory synaptic transmission both in vitro and in vivo. Thus, we conclude that the optogenetic effects we observed in Vgat-Cre mice throughout this study can indeed be attributed to the activity of inhibitory synaptic transmission.

Further, the Vgat^{Cre} line that was used in our study is publicly available on JAX, and has been extensively characterized anatomically, electrophysiologically, and molecular biologically. References include the following publications in high-profile Journals:

Vong L et al., 2011, Neuron. - Colocalization of Vgat-Cre reporter with Vgat mRNA and known anatomical locations of inhibitory neurons.

Mahoney CE et al., 2017, J Neurosci - Colocalization of Vgat-Cre reporter with Vgat mRNA

Garfield AS et al., 2016, Nat Neurosci - Electrophysiological characterization of the inhibitory influence of Vgat-ChR2 stimulation on postsynaptic neurons.

Seo DO et al., 2016, J Neurosci - Colocalization of Vgat-Cre reporter with GAD65/67 immunostaining and optogenetic effects blocked pharmacologically.

Also used in: Geerling et al., 2016; Martin et al., 2014; Wu and Tollkhun, 2017; Koga et al., 2017; Geerling et al., 2017... and more.

We now cite Vong et al., 2011 in the text. However, due to reference limits set by the Journal, we cannot cite all these studies in the manuscript. As detailed below, we also added an intracellular recording of a Vgat- neuron that is hyperpolarized during light stimulation.

2. Conclusion that activation increased respiratory following activation of inhibitory is not due to post-inhibitory rebound. On page 11 the authors state that the rapid bursting observed following optogenetic activation of Vgat-cre neurons in slice is not due to post-inhibitory rebound. However, the control experiment for this, Figure 3H, is insufficient to make this conclusion. Given that the length of the refractory period is dictated by the extent and length of depolarization, Figure 3H is measuring the probability of post-inhibitory rebound following a normal complete burst. While in comparison, Figure 3E is a prematurely terminated burst after activation of the preBötC Vgat-cre neurons. Therefore, it remains a likely possibility that the refractory period is indeed much shorter after a terminated burst, Figure 3E. The experiment in Figure 3H should be repeated and the refractory period should be measured following a terminated burst as in 3E.

The reviewer is correct in stating that the refractory period will be much shorter after a terminated burst, and we apologize for not making this clear in our original submission. Indeed, our data suggest that the rhythm speeds up because the refractory period is shortened or eliminated. In Fig. 5d we demonstrate that if the preceding burst is not terminated, mechanisms of postinhibitory rebound are constrained to the same refractory period as activation of Dbx1 neurons. Thus, we conclude that mechanisms of postinhibitory rebound per se cannot overcome the refractory period to dramatically increase breathing frequency. We appreciate the reviewers comment and have tried to clarify this in the revised text to avoid any further confusion. We also now include example intracellular recordings of a Vgat- neuron during postinhibitory rebound showing reduced spiking during the refractory period (Fig. 5d).

3. Conclusion that Dbx1-derived neurons are distinct from Vglut2 neurons. The manuscript eludes to a population of excitatory neurons that are distinct from the Dbx1-derived excitatory neurons and that these neurons have distinct roles in breathing (Figures 5-7). However, critically, they have labeled Dbx1-derived neurons by injecting tamoxifen into pregnant dams at e10.5 which will induce recombination via the Dbx1-creER transgene. Kottick et al. (2017). Fate mapping neurons and glia derived from Dbx1-expressing progenitors in mouse preBötzing complex. Physiol. Rep. recently reported that Dbx1-expressing neurons can

be labeled by tamoxifen from e8.5 to e11.5. Therefore, by only injecting tamoxifen at e10.5, this work has failed to completely label Dbx1 derived preBötC neurons. Therefore, the conclusion that they are functionally distinct is incorrect since the Vglut2-cre transgene should fully label all Dbx1-derived excitatory neurons. In order to make this conclusion, the authors should repeat their in vivo experiments with the Dbx1-cre transgene to more completely label Dbx1-derived neurons. Furthermore, they should also repeat the in vitro experiments with Vglut2-cre.

We fully agree with the reviewer that the overlap between Vglut2 and Dbx1 neurons is an important issue. It is our understanding that not all Vglut2+ neurons in the preBötC are Dbx1-derived, but that nearly all Dbx1 neurons in preBötC express Vglut2. This was shown previously in two landmark papers: Bouvier et al., 2010, and Gray et al., 2010, where Dbx1LacZ/+ mice were used, which is expected to label all Dbx1 derived cells. These studies suggest that there is a population of Vglut2 neurons in preBötC that are distinct from Dbx1 neurons.

We also agree with the reviewer that Dbx1-Cre ERT2 will not completely label Dbx1-derived neurons, as shown in the elegant paper by Kottick et al., 2017. Hence, to make sure that we capture all glutamatergic neurons, we report experiments performed with Dbx1-CreERT2 and the Vglut2-Cre line, which completely labels glutamatergic neurons.

What we observe is that light stimulation in Vglut2-ChR2 mice has a larger effect on frequency than stimulation in Dbx1-ChR2 mice. However, it is important to note that this difference was only observed with vagal feedback intact. By contrast, in vitro and in vagotomized mice stimulation of Vglut2-ChR2 and Dbx1-ChR2 had the same effect. The differences were restricted to burst frequency, but not amplitude, and there was only a difference during expiratory stimulation, but not inspiratory stimulation. These critical observations are difficult to reconcile if Dbx1-CreERT2 and Vglut2-Cre label the same functional population. Therefore, we conclude that vagal feedback differentially influences neurons labeled by Dbx1-CreERT2 and Vglut2-Cre.

However, we agree with the reviewer, that it is possible that this functional distinction occurs among Dbx1-derived neurons since they are incompletely labeled by Dbx1-CreERT2. But, this would imply that Dbx1 neurons may have functional variation depending on their "birth date" (i.e. timing of tamoxifen administration). We have revised our interpretation and the text to reflect these considerations accordingly.

We performed additional experiments, and now include an examination of the refractory period in Vglut2-ChR2 slices in vitro. However, we feel that our conclusions will not change if we would repeat all of the in vivo experiments using the constitutive Dbx1-Cre transgene. Whether stimulation in this mouse line replicates our observations in Dbx1-CreERT2 or Vglut2 vagus-intact mice, the results would have little impact on our overall conclusion, i.e. that excitatory mechanisms are restrained by the refractory period, which is the major thrust of the paper.

4. Conclusion that refractory period of Dbx1 neurons determines burst rate in vitro. Figures 1 and 2 show a nice correlation that the Dbx1 neuron refractory period correlates with the probability of bursting. However, the authors define the refractory period by activating only a subset of Dbx1-derived neurons. This may not be the true refractory period since they have incompletely labeled Dbx1-derived neurons (discussed above). The in vitro refractory period (Figure 2C) should be determined with the Vglut2-cre line. As in Figure 7Ai, it may be possible for bursting to occur earlier in this transgenic line, which would force the authors to reinterpret the endogenous bursting rate and the refractory period.

In our initial studies, we used the Dbx1-CreERT2 line to replicate the elegant findings of Kottick and Del Negro, 2015 who first identified the Dbx1 refractory period using Dbx1-CreERT2 mice given tamoxifen at E10.5. Further, Dbx1 neurons from Dbx1-CreERT2 mice given tamoxifen at E10.5 are the critical neurons for initiating bursts (Wang et al., 2014) and therefore we expected that the refractory period specific to these neurons would be most relevant for the delay before the next burst is initiated, thereby altering frequency.

However, to address the reviewer's suggestion, we have now repeated this series of experiments in Vglut2-Cre mice and found similar results (Fig. 1a). This is not necessarily surprising since, in our original work, we also quantified the refractory period for Dbx1 in vivo and compared it to the refractory period during Vglut2 stimulation (which will label nearly all Dbx1 neurons), and find that the refractory period is the same for Dbx1 and Vglut2 stimulation in vagotomized mice (slower rhythm lacking sensory feedback inhibition like the slice preparation).

Secondary concerns:

1. Determining if prematurely terminated bursts in vitro are transmitted through the hypoglossal nerve. In Figure 3E the preBötC activity is measured with an extracellular electrode. This electrode will pick up the ChR2 induced depolarization of Vgat-cre derived neurons. Therefore, the preBötC burst should be measured by its downstream surrogate, cranial nerve 12 rootlet. It is important to know that the prematurely terminated burst is actually leading to a motor output of a breath.

This requested experiment was performed in vivo where we found nearly identical results. Specifically, Vgat stimulation triggered off XII activity in vivo terminated inspiratory bursts and advanced the next cycle. Terminated bursts continued to be transmitted to motor output as observed in XII nerve recordings in vivo.

2. Characterization of Dbx1 depolarization during expiration. In Figure 3F the authors activate Dbx1-derived neurons during expiration. According to their model, if Dbx1 neurons never recovery from the refractory period, then a burst should not occur. How is it that Dbx1 neurons recovery from the refractory period while they are depolarized by ChR2 during all of expiration. Whole cell recordings of Dbx1 neurons should be performed during the light pulse to characterize the extent of depolarization that occurs during a prolonged light pulse through expiration.

We agree with the reviewer, that if the neurons were not able to recover from the refractory period a subsequent burst should not occur. But we do not expect that activation of these neurons during expiration would “lock” them in a refractory mechanism. To the contrary, we suggest that stimulation during expiration accelerates burst onset following the refractory period by increasing the rate of spread of excitation through the network (percolation), whereas stimulation during inspiration exaggerates the refractory period and delays the onset of the next burst.

In Fig.2e we perform cell-specific current injections in isolated Dbx1 neurons at various times following an artificial “burst” to demonstrate how these neurons respond to depolarization during the refractory period. In addition, as the reviewer suggests, we now include an example trace of a whole cell recording of a Dbx1 neuron during light pulses, showing reduced excitation during the refractory period (Supplementary Fig. 2a).

3. Please add quantification for Figure 1F and 1G.

As suggested by the reviewer, we have revised the Figures and added the requested quantified data.

4. Please increase the N for Figure 2A and B. The spread after adding gabazine is quite large and should be more convincing.

As suggested by the reviewer, we have performed additional experiments and now include n=7 and n=11 for changes in drive potential area and burst amplitude, respectively. This is reflected in the revised figure and legend.

5. The text should not state that Zsfgreen is GFP. These are different proteins. Perhaps they could call Zsfgreen a reporter instead.

We thank the reviewer for this comment. The text has been corrected accordingly in the revised manuscript.

Reviewers' comments:

Reviewer #1 (Remarks to the Author):

The authors have responded to my comments in an appropriate way, and I have no further questions.

Reviewer #2 (Remarks to the Author):

Let me state that the authors have done a heroic job responding to my (very lengthy) critique after the first round. They addressed my concerns very well, but I have a few further issues that I think need clarification. I hope in addressing them the authors will further improve this important and interesting study (I mean these comments to be helpful).

MAJOR

The diagram in Fig. 1a is quite helpful in organizing the composition of the preBötC.

Lines 469-480: Here the authors address the caveat that more profound effects on the phase-response curve in Vglut2-ChR2 mice, compared to Dbx1-ChR2 mice, could be due to Vglut2-expressing axon terminals in the preBötC. One expects that ChR2-expressing axon terminals would be more abundant in Vglut2-ChR2 mice compared to Dbx1-ChR2 mice. The authors acknowledge the caveat, which is something this reviewer encouraged in the first round. I think this issue is very important because it pertains to the whole organization of the preBötC as the authors show so well in Fig. 1a. If the differential (i.e., stronger) effects of stimulating Vglut2-ChR2 mice is because the non-Dbx1 excitatory neuron population is large, then that implies that the Dbx1 rhythmogenic core is smaller than most consider it to be (and thus that non-Dbx1 excitatory neurons are more numerous and influential than previously appreciated). If, however, the differential effects of stimulating Vglut-ChR2 mice is because of an enormous field of excitatory ChR2-expressing axon terminals in the preBötC that respond to light, then the "green" rectangle in the top right of Fig. 1a is likely small, and the relative size of the Dbx1 excitatory core is larger than the "green+blue" rectangle at the upper left. At this point we really don't know the relative sizes of the Dbx1 and non-Dbx1 excitatory populations in the preBötC. The present data may lead to speculation that the non-Dbx1 excitatory core is large since the phase response curve is much more pronounced in that mouse model (their Fig. 9c). I am sure the authors appreciate the importance of this point.

MEDIUM

Since the first round, another paper on the role of inhibition in respiration was recently published (Cregg et al. PNAS doi: 10.1073/pnas.1711536114) which addresses some of the same points, but with far less rigor and with no slice or whole-animal experiments. That paper speculates that post-inhibitory rebound (PIR) is rhythmogenic, but the authors'

present data (compare Fig. 5d in vitro to 9d in vivo) shows that PIR might appear to be rhythmogenic in vitro, but that idea does not hold up in whole animal experiments, so is likely wrong. In my view, that should be pointed out so that the high visibility of the other paper (Cregg et al.) does not mislead too many investigators in the field.

MINOR COMMENTS

In the Abstract (line 32) and Introduction (line 76) I do not think refractory period needs to be in quotes; the authors are quite literally referring to an actual refractory period.

An oxford comma may be helpful in line 49 (after locomotion).

When referring to the role of Dbx1 neurons in respiratory rhythmogenesis (lines 109-110) I would respectfully suggest one additional reference for the "(e.g., 26,27,29,33,34)" and that is: Vann et al. Transient suppression of Dbx1 preBötzinger interneurons disrupts breathing in adult mice. PLoS ONE 11, e0162418, 2016.

Line 122: A dash does not seem necessary in "light-evoking".

Line 142: the first reference to a supplemental figure is here, referring to Supplementary Figure 2b. What about Supplementary Figure 1 and 2a?

Why is Figure 3d (line 162) presented before 3c (line 168)? Perhaps the panel order should be swapped?

Line 168: Supplementary Figure 1 (panel b) is used after Supplementary Figure 2, and no reference is made to Supplementary Figure 1a until line 231. Nor is there a citation to Supplementary Figure 2a. In general, I think the authors should present the figures and figure panels in a sequential order, including supplementary figures. If supplementary (or normal) figure panels are not used in the narrative, perhaps they are not necessary for paper? I strongly urge the authors to reorganize the order of the figure panels, normal and supplemental so that the narrative flow in the manuscript closely corresponds to the number order of the illustrations throughout. If not, you frustrate the otherwise keen reader.

Figure 3d: Both y-axes should specify "cumulative probability" (not just the top one illustrating refractory period) and in the figure legend (line 995) the units are not specified for the bins related to frequency, although I presume it is "Hz". Please add the unit in the figure legend.

Lines 249-250: The syntax is odd "Similar to20, but see21, ..." Could the authors just refer to the authors, Janczewski et al. and Marchenko et al.?

Line 329: The authors refer to "vagus-intact mice" citing Fig. 9a-b, but Fig. 9b is vagotomized conditions. Either this is a typo or the narrative description needs to change.

Line 359: It is not immediately clear how the bar chart in Figure 5c agrees with the irregularity point in Figure 10a,b referred to here. After considering this for quite some time, I think the authors mean that the irregularity in vivo (10a,b) is quite similar to the broad distribution of individual data points under the inspiratory stimulation of Vgat2-ChR2 in vitro. Given the purple bars, and small grey data points, it's very difficult for the reader to make out those points and make sense of their disparity. Or perhaps I don't know what they are getting at by line 359 and the comparison between 10a,b and 5c. I recommend some type of clarification either visual display or by writing.

Lines 448-449: regarding the point that different populations of inhibitory neurons play different roles, I recommend citing Shao & Feldman J Neurophysiol 77: 1853-60, 1997, where they showed that glycinergic inhibition is operational only during the expiratory phase in preBötC neurons whereas GABAergic and glycinergic mechanisms modulate neuronal excitability across the entire cycle.

Reviewer #3 (Remarks to the Author):

I am satisfied with the answers to my questions.

Reviewer #4 (Remarks to the Author):

The reformat and additional experiments have significantly improved the flow of the manuscript. The story is now much easier to read and digest leading to a better understanding. Great job.

Of the comments during the first round of revision, only these two concerns remain to be satisfactorily addressed:

3. Conclusion that Dbx1-derived neurons are distinct from Vglut2 neurons. The manuscript eludes to a population of excitatory neurons that are distinct from the Dbx1-derived excitatory neurons and that these neurons have distinct roles in breathing (Figures 5-7). However, critically, they have labeled Dbx1-derived neurons by injecting tamoxifen into pregnant dams at e10.5 which will induce recombination via the Dbx1-creER transgene. Kottick et al. (2017). Fate mapping neurons and glia derived from Dbx1-expressing progenitors in mouse preBötzing complex. Physiol. Rep. recently reported that Dbx1-expressing neurons can be labeled by tamoxifen from e8.5 to e11.5. Therefore, by only injecting tamoxifen at e10.5, this work has failed to completely label Dbx1-derived preBötC neurons. Therefore, the conclusion that they are functionally distinct is incorrect since the Vglut2-cre transgene should fully label all Dbx1-derived excitatory neurons. In order to make this conclusion, the authors should repeat their in vivo experiments with the Dbx1-cre transgene to more completely label Dbx1-derived neurons. Furthermore, they should also repeat the in vitro experiments with Vglut2-cre.

We fully agree with the reviewer that the overlap between Vglut2 and Dbx1 neurons is an

important issue. It is our understanding that not all Vglut2+ neurons in the preBötC are Dbx1-derived, but that nearly all Dbx1 neurons in preBötC express Vglut2. This was shown previously in two landmark papers: Bouvier et al., 2010, and Gray et al., 2010, where Dbx1LacZ/+ mice were used, which is expected to label all Dbx1 derived cells. These studies suggest that there is a population of Vglut2 neurons in preBötC that are distinct from Dbx1 neurons.

We also agree with the reviewer that Dbx1-Cre ERT2 will not completely label Dbx1-derived neurons, as shown in the elegant paper by Kottick et al., 2017. Hence, to make sure that we capture all glutamatergic neurons, we report experiments performed with Dbx1-CreERT2 and the Vglut2-Cre line, which completely labels glutamatergic neurons.

What we observe is that light stimulation in Vglut2-ChR2 mice has a larger effect on frequency than stimulation in Dbx1-ChR2 mice. However, it is important to note that this difference was only observed with vagal feedback intact. By contrast, in vitro and in vagotomized mice stimulation of Vglut2-ChR2 and Dbx1-ChR2 had the same effect. The differences were restricted to burst frequency, but not amplitude, and there was only a difference during expiratory stimulation, but not inspiratory stimulation. These critical observations are difficult to reconcile if Dbx1-CreERT2 and Vglut2-Cre label the same functional population. Therefore, we conclude that vagal feedback differentially influences neurons labeled by Dbx1-CreERT2 and Vglut2-Cre.

However, we agree with the reviewer, that it is possible that this functional distinction occurs among Dbx1-derived neurons since they are incompletely labeled by Dbx1-CreERT2. But, this would imply that Dbx1 neurons may have functional variation depending on their "birth date" (i.e. timing of tamoxifen administration). We have revised our interpretation and the text to reflect these considerations accordingly.

We performed additional experiments, and now include an examination of the refractory period in Vglut2-ChR2 slices in vitro. However, we feel that our conclusions will not change if we would repeat all of the in vivo experiments using the constitutive Dbx1-Cre transgene. Whether stimulation in this mouse line replicates our observations in Dbx1-CreERT2 or Vglut2 vagus-intact mice, the results would have little impact on our overall conclusion, i.e. that excitatory mechanisms are restrained by the refractory period, which is the major thrust of the paper. 

The clarification in the text (p. 4-5) that the Dbx1 transgene used in these studies is the Dbx1-creER instead of the Dbx1-cre clarifies to the reader a limitation when claiming that the subsequent studies represent manipulations to the entire Dbx1 neural lineage. However, the conclusion that there is a distinct Vglut2 neural population that receives specific vagal input is remains speculative (Fig 10e, and p. 26) and still needs to be addressed.

First, as noted in the first round of revision, it remains possible that a manipulation of the entire Dbx1-lineage could mimic the Vglut-cre experiments.

Second, it is contradictory for stimulation of Vglut2 neurons to overcome the refractory period of Dbx1 neurons. If Dbx1 neurons are required for a breath (cited within this paper), then their longer refractory period should be the rate limiting step. For example, in order for

the optogenetic activation of Vglut2 neurons to increase the respiratory frequency beyond the increase observed when activating Dbx1 neurons (Fig 7c-d), then the Vglut2 stimulation must somehow shorten the Dbx1 refractory period. Based on the authors models and the fact that the refractory period is cell autonomous (Fig 2 and Suppl. Fig 2), it seems impossible how this could happen. In all likelihood, and more consistent with the connection between tidal volume and refractory length, the stimulation of Vglut2 neurons leads to a smaller breath (Fig 7c), and thus a shorter refractory period, allowing for the frequency to also slightly increase (Fig 7c). This model is more parsimonious than the current one.

2. Characterization of Dbx1 depolarization during expiration. In Figure 3F the authors activate Dbx1-derived neurons during expiration. According to their model, if Dbx1 neurons never recovery from the refractory period, then a burst should not occur. How is it that Dbx1 neurons recovery from the refractory period while they are depolarized by ChR2 during all of expiration. Whole cell recordings of Dbx1 neurons should be performed during the light pulse to characterize the extent of depolarization that occurs during a prolonged light pulse through expiration.

We agree with the reviewer, that if the neurons were not able to recover from the refractory period a subsequent burst should not occur. But we do not expect that activation of these neurons during expiration would "lock" them in a refractory mechanism. To the contrary, we suggest that stimulation during expiration accelerates burst onset following the refractory period by increasing the rate of spread of excitation through the network (percolation), whereas stimulation during inspiration exaggerates the refractory period and delays the onset of the next burst. **

In Fig.2e we perform cell-specific current injections in isolated Dbx1 neurons at various times following an artificial "burst" to demonstrate how these neurons respond to depolarization during the refractory period. In addition, as the reviewer suggests, we now include an example trace of a whole cell recording of a Dbx1 neuron during light pulses, showing reduced excitation during the refractory period (Supplementary Fig. 2a). The whole cell recordings in Figure 2e and supplemental Figure 2 suggest that the refractory period is an intrinsic property of the neurons which originates in the biophysical properties of their ion channels. The refractory period does not appear to be a property of the network as suggested above. Therefore, it seems that sustained depolarizing ChR2 stimulation during expiration, if sufficiently depolarizing, should maintain neurons in a state in which the voltage gated channels, like the sodium channels, stay desensitized. The interesting model of this paper and data supporting a refractory period requires a period of hyperpolarization to re-establish a state in which the next breath can be triggered. Thus, with the idea that ChR2 is depolarizing the neurons throughout expiration, it appears to contradict the model. In all likelihood, ChR2 itself is inactivating during this prolonged stimulation and therefore allowing for the hyperpolarized refractory period to take place. The only way to understand this would be to perform whole cell recordings of the Dbx1 neurons throughout the prolonged expiratory light stimulation. Although this may not be the most critical element of the paper, it is an important contradiction to a model implicating an cell autonomous refractory period defining expiratory length and should be addressed.

Reviewer #2 (Remarks to the Author):

Let me state that the authors have done a heroic job responding to my (very lengthy) critique after the first round. They addressed my concerns very well, but I have a few further issues that I think need clarification. I hope in addressing them the authors will further improve this important and interesting study (I mean these comments to be helpful).

We thank the reviewer for their kind comments and valued advice on our manuscript.

MAJOR

The diagram in Fig. 1a is quite helpful in organizing the composition of the preBötC.

Lines 469-480: Here the authors address the caveat that more profound effects on the phase-response curve in Vglut2-ChR2 mice, compared to Dbx1-ChR2 mice, could be due to Vglut2-expressing axon terminals in the preBötC. One expects that ChR2-expressing axon terminals would be more abundant in Vglut2-ChR2 mice compared to Dbx1-ChR2 mice. The authors acknowledge the caveat, which is something this reviewer encouraged in the first round. I think this issue is very important because it pertains to the whole organization of the preBötC as the authors show so well in Fig. 1a. If the differential (i.e., stronger) effects of stimulating Vglut2-ChR2 mice is because the non-Dbx1 excitatory neuron population is large, then that implies that the Dbx1 rhythmogenic core is smaller than most consider it to be (and thus that non-Dbx1 excitatory neurons are more numerous and influential than previously appreciated). If, however, the differential effects of stimulating Vglut-ChR2 mice is because of an enormous field of excitatory ChR2-expressing axon terminals in the preBötC that respond to light, then the “green” rectangle in the top right of Fig. 1a is likely small, and the relative size of the Dbx1 excitatory core is larger than the “green+blue” rectangle at the upper left. At this point we really don’t know the relative sizes of the Dbx1 and non-Dbx1 excitatory populations in the preBötC. The present data may lead to speculation that the non-Dbx1 excitatory core is large since the phase response curve is much more pronounced in that mouse model (their Fig. 9c). I am sure the authors appreciate the importance of this point.

We certainly appreciate the reviewers point.

Indeed, we don’t want our results to “lead to speculation that the non-Dbx1 excitatory core is large since the phase response curve is much more pronounced in that mouse model”. On the contrary, what we speculate in the text is that non-Dbx1 neurons as less active and therefore likely play less of a functional role in rhythmogenesis than Dbx1 neurons. However, precisely because non-Dbx1 neurons are less active, they have a shorter refractory period when evoked with ChR2.

The diagram in Fig. 1a is based off prior histological studies (Bouvier et al., 2010; Gray et al., 2013, 2010; Winter et al., 2009) that have quantified the relative numbers of Dbx1 and non-Dbx1 glutamatergic neurons and inhibitory neurons. (We have added these references to the figure legend). We did not repeat this work. However, we believe our functional results support this preBotC architecture. As the reviewer points out, the number of cell bodies contained within the preBotC (diagramed in Fig.1a) does not necessarily reflect the number of synapses innervating the area and therefore the corresponding functional effect of ChR2 stimulation (since ChR2 is expected to activate neurons terminals).

Also, we agree with the reviewer that very little is known with regard to the role of non-Dbx1 glutamatergic neurons (or whether their role is easily distinguishable from Dbx1 neurons). Whether these specific neurons are necessary or sufficient for generating inspiratory bursts remains unknown (Since it is

difficult to specifically target the non-Dbx1 glutamatergic population). The synaptic inputs to Dbx1 neurons are also not known – they could receive the majority of their synaptic input from other Dbx1 neurons, or the majority from other non-Dbx1 synapses. Dissecting this functional connectivity in detail will surely be a very interesting area of investigation, but is beyond the scope of this study.

As mentioned in our initial response, we do not intend for the differences observed between Dbx1 and Vglut2 stimulation to distract from the major point of the paper: that ALL excitatory mechanisms are limited by refractory mechanisms and inhibition is required to modulate these refractory properties. Thus, we have tried to further soften this point in the text while including all possible explanations of the data with the simplest explanation (in our opinion) outlined in our summary model (Fig. 10e).

MEDIUM

Since the first round, another paper on the role of inhibition in respiration was recently published (Cregg et al. PNAS doi: 10.1073/pnas.1711536114) which addresses some of the same points, but with far less rigor and with no slice or whole-animal experiments. That paper speculates that post-inhibitory rebound (PIR) is rhythmogenic, but the authors' present data (compare Fig. 5d in vitro to 9d in vivo) shows that PIR might appear to be rhythmogenic in vitro, but that idea does not hold up in whole animal experiments, so is likely wrong. In my view, that should be pointed out so that the high visibility of the other paper (Cregg et al.) does not mislead too many investigators in the field.

We agree with the reviewer and now state in the text:

“... in contrast to in vitro preparations, stimulation of inhibitory neurons during expiration did not evoke postinhibitory rebound in vivo, calling into question models of respiratory rhythm generation implicating postinhibitory rebound as a driver of rapid breathing based on a very slow non-physiological rhythm generated in situ (Cregg et al., 2017).”

“However, our results suggest that postinhibitory rebound is unlikely to be a driver of rapid breathing (Cregg et al., 2017) under physiological conditions, since rebound from inhibition does not overcome the refractory period (Fig. 5d) and does not generate inspiratory bursts in vivo (Fig. 9).”

MINOR COMMENTS

In the Abstract (line 32) and Introduction (line 76) I do not think refractory period needs to be in quotes; the authors are quite literally referring to an actual refractory period. **Corrected**

An oxford comma may be helpful in line 49 (after locomotion). **Corrected**

When referring to the role of Dbx1 neurons in respiratory rhythmogenesis (lines 109-110) I would respectfully suggest one additional reference for the “(e.g., 26,27,29,33,34)” and that is: Vann et al. Transient suppression of Dbx1 preBöttinger interneurons disrupts breathing in adult mice. PLoS ONE 11, e0162418, 2016. **Citation added as requested**

Line 122: A dash does not seem necessary in “light-evoking”. **Corrected**

Line 142: the first reference to a supplemental figure is here, referring to Supplementary Figure 2b. What about Supplementary Figure 1 and 2a? **We now reference all supplemental figure panels in the text.**

Why is Figure 3d (line 162) presented before 3c (line 168)? Perhaps the panel order should be swapped?

Line 168: Supplementary Figure 1 (panel b) is used after Supplementary Figure 2, and no reference is made to Supplementary Figure 1a until line 231. Nor is there a citation to Supplementary Figure 2a. In general, I think the authors should present the figures and figure panels in a sequential order, including supplementary figures. If supplementary (or normal) figure panels are not used in the narrative, perhaps they are not necessary for paper? I strongly urge the authors to reorganize the order of the figure panels, normal and supplemental so that the narrative flow in the manuscript closely corresponds to the number order of the illustrations throughout. If not, you frustrate the otherwise keen reader.

We thank the reviewer for pointing this out and have tried to improve the figure citations and the order of the figure panels where possible (e.g. Fig. 3 has been rearranged to match the order of the text). However, as I'm sure the reviewer can appreciate, this is not always possible while staying within the figure limit, while making figure legends proceed in a logical order, and while making the figures aesthetically pleasing. For example, Supplementary Fig 1a and 1b correspond to different sections of the text. But they are the same type of analysis and fit together nicely when viewed as one figure. In our opinion, this is a better option than spitting the figure into two individual figures (there are already 9) with one graph each.

Figure 3d: Both y-axes should specify “cumulative probability” (not just the top one illustrating refractory period) and in the figure legend (line 995) the units are not specified for the bins related to frequency, although I presume it is “Hz”. Please add the unit in the figure legend. **The x-axis units are seconds post endogenous burst as shown in the figure. We now also include this information in the legend.**

Lines 249-250: The syntax is odd “Similar to20, but see21, ...” Could the authors just refer to the authors, Janczewski et al. and Marchenko et al.? **Corrected as requested**

Line 329: The authors refer to “vagus-intact mice” citing Fig. 9a-b, but Fig. 9b is vagotomized conditions. Either this is a typo or the narrative description needs to change. **Corrected**

Line 359: It is not immediately clear how the bar chart in Figure 5c agrees with the irregularity point in Figure 10a,b referred to here. After considering this for quite some time, I think the authors mean that the irregularity in vivo (10a,b) is quite similar to the broad distribution of individual data points under the inspiratory stimulation of Vgat2-ChR2 in vitro. Given the purple bars, and small grey data points, it's very difficult for the reader to make out those points and make sense of their disparity. Or perhaps I don't know what they are getting at by line 359 and the comparison between 10a,b and 5c. I recommend some type of clarification either visual display or by writing.

Thanks for pointing this out. The irregularity during unilateral stimulation is not displayed in Fig 5c. Instead we refer to it in the text “bursting was more irregular and slower on average during unilateral stimulation”. Thus, we no longer reference this figure here, but still would like to point out the similarity between the in vitro and in vivo condition.

Lines 448-449: regarding the point that different populations of inhibitory neurons play different roles, I recommend citing Shao & Feldman J Neurophysiol 77: 1853-60, 1997, where they showed that glycinergic inhibition is operational only during the expiratory phase in preBötC neurons whereas GABAergic and glycinergic mechanisms modulate neuronal excitability across the entire cycle.

We now include this citation as requested.

Reviewer #4 (Remarks to the Author):

The reformat and additional experiments have significantly improved the flow of the manuscript. The story is now much easier read and digest leading to a better to understanding. Great job.

We thank the reviewer for their assistance in improving our manuscript.

Of the comments during the first round of revision, only these two concerns remain to be satisfactorily addressed:

3. Conclusion that Dbx1-derived neurons are distinct from Vglut2 neurons. The manuscript eludes to a population of excitatory neurons that are distinct from the Dbx1-derived excitatory neurons and that these neurons have distinct roles in breathing (Figures 5-7). However, critically, they have labeled Dbx1-derived neurons by injecting tamoxifen into pregnant dams at e10.5 which will induce recombination via the Dbx1-creER transgene. Kottick et al. (2017). Fate mapping neurons and glia derived from Dbx1-expressing progenitors in mouse preBötzing complex. Physiol. Rep. recently reported that Dbx1-expressing neurons can be labeled by tamoxifen from e8.5 to e11.5. Therefore, by only injecting tamoxifen at e10.5, this work has failed to completely label Dbx1 derived preBötC neurons. Therefore, the conclusion that they are functionally distinct is incorrect since the Vglut2-cre transgene should fully label all Dbx1-derived excitatory neurons. In order to make this conclusion, the authors should repeat their in vivo experiments with the Dbx1-cre transgene to more completely label Dbx1-derived neurons. Furthermore, they should also repeat the in vitro experiments with Vglut2-cre.

We fully agree with the reviewer that the overlap between Vglut2 and Dbx1 neurons is an important issue. It is our understanding that not all Vglut2+ neurons in the preBötC are Dbx1-derived, but that nearly all Dbx1 neurons in preBötC express Vglut2. This was shown previously in two landmark papers: Bouvier et al., 2010, and Gray et al., 2010, where Dbx1LacZ/+ mice were used, which is expected to label all Dbx1 derived cells. These studies suggest that there is a population of Vglut2 neurons in preBötC that are distinct from Dbx1 neurons.

We also agree with the reviewer that Dbx1-Cre ERT2 will not completely label Dbx1-derived neurons, as shown in the elegant paper by Kottick et al., 2017. Hence, to make sure that we capture all glutamatergic neurons, we report experiments performed with Dbx1-CreERT2 and the Vglut2-Cre line, which completely labels glutamatergic neurons.

What we observe is that light stimulation in Vglut2-ChR2 mice has a larger effect on frequency than stimulation in Dbx1-ChR2 mice. However, it is important to note that this difference was only observed with vagal feedback intact. By contrast, in vitro and in vagotomized mice stimulation of Vglut2-ChR2 and Dbx1-ChR2 had the same effect. The differences were restricted to burst frequency, but not amplitude, and there

was only a difference during expiratory stimulation, but not inspiratory stimulation. These critical observations are difficult to reconcile if Dbx1-CreERT2 and Vglut2-Cre label the same functional population. Therefore, we conclude that vagal feedback differentially influences neurons labeled by Dbx1-CreERT2 and Vglut2-Cre.

However, we agree with the reviewer, that it is possible that this functional distinction occurs among Dbx1-derived neurons since they are incompletely labeled by Dbx1-CreERT2. But, this would imply that Dbx1 neurons may have functional variation depending on their “birth date” (i.e. timing of tamoxifen administration). We have revised our interpretation and the text to reflect these considerations accordingly. We performed additional experiments, and now include an examination of the refractory period in Vglut2-ChR2 slices in vitro. However, we feel that our conclusions will not change if we would repeat all of the in vivo experiments using the constitutive Dbx1-Cre transgene. Whether stimulation in this mouse line replicates our observations in Dbx1-CreERT2 or Vglut2 vagus-intact mice, the results would have little impact on our overall conclusion, i.e. that excitatory mechanisms are restrained by the refractory period, which is the major thrust of the paper.

The clarification in the text (p. 4-5) that the Dbx1 transgene used in these studies is the Dbx1-creER instead of the Dbx1-cre clarifies to the reader a limitation when claiming that the subsequent studies represent manipulations to the entire Dbx1 neural lineage. However, the conclusion that there is a distinct Vglut2 neural population that receives specific vagal input is remains speculative (Fig 10e, and p. 26) and still needs to be addressed.

This is certainly a limitation of Dbx1-CreERT2. Accordingly, we are very careful NOT to make the claim that our manipulations activate the entire Dbx1 lineage. Indeed, as mentioned in our initial response, we do not intend for the differences observed between Dbx1 and Vglut2 stimulation to distract from the major point of the paper: that ALL excitatory mechanisms are limited by refractory mechanisms and inhibition is required to modulate these refractory properties. Thus, we have tried to further soften this point in the text while including all possible explanations of the data with the simplest explanation (in our opinion) outlined in our summary model (Fig. 10e).

We now specify in the text that this is a working model and state “Although this working model of the preBotC best fits our results, the proposed connectivity lacks direct evidence, which will be an important avenue for future investigations.”

See below for detailed explanation.

First, as noted in the first round of revision, it remains possible that a manipulation of the entire Dbx1-lineage could mimic the Vglut-cre experiments.

We acknowledge this possibility. Indeed, when performing these experiments, this was our initial interpretation. However, we later realized that there are significant discrepancies between this interpretation and our results:

1) It does not explain why the difference between Dbx1 and Vglut2 stimulation is specific to the vagus-intact condition. The differential frequency responses observed between Dbx1 and Vglut2 stimulation are dependent on vagal feedback (Fig.7-9). We also did not observe a difference between Dbx1 and Vglut2 stimulation in vitro. If the difference between Dbx1 and Vglut2 stimulation is solely due to stimulating less Dbx1 neurons in one condition vs. the other, we would expect that there would continue to be a difference

between Dbx1 and Vglut2 stimulation in vagotomized mice, and we would expect a difference in the slice preparation.

2) It does not explain why the difference between Dbx1 and Vglut2 stimulation is specific to the expiratory phase. If stimulations are targeted to the inspiratory phase, there is no difference between Dbx1 and Vglut2 stimulation in vagus-intact OR vagotomized mice (Fig. 8a,c). If the difference between Dbx1 and Vglut2 stimulation is solely due to stimulating less Dbx1 neurons in one condition vs. the other, we would expect larger effects of Vglut2 stimulation regardless of phase. I.e. why doesn't Vglut2 stimulation during inspiration increase amplitude and delay the next burst more than Dbx1 stimulation?

3) It does not explain why the difference between Dbx1 and Vglut2 stimulation is specific to burst frequency but not amplitude. If the difference between Dbx1 and Vglut2 stimulation is solely due to stimulating less Dbx1 neurons in one condition vs. the other, we would expect the amplitude response of Dbx1 stimulation to also be smaller than the amplitude response of Vglut2 stimulations. However, this is not what we observed.

Thus, although we try to present all possibilities in the text, we believe the conclusion that vagal feedback differentially targets Dbx1 and Vglut2 populations is the simplest and most logical explanation in our opinion.

Second, it is contradictory for stimulation of Vglut2 neurons to overcome the refractory period of Dbx1 neurons. If Dbx1 neurons are required for a breath (cited within this paper), then their longer refractory period should be the rate limiting step. For example, in order for the optogenetic activation of Vglut2 neurons to increase the respiratory frequency beyond the increase observed when activating Dbx1 neurons (Fig 7c-d), then the Vglut2 stimulation must somehow shorten the Dbx1 refractory period. Based on the authors models and the fact that the refractory period is cell autonomous (Fig 2 and Suppl. Fig 2), it seems impossible how this could happen. In all likelihood, and more consistent with the connection between tidal volume and refractory length, the stimulation of Vglut2 neurons leads to a smaller breath (Fig 7c), and thus a shorter refractory period, allowing for the frequency to also slightly increase (Fig 7c). This model is more parsimonious than the current one.

There is substantial evidence that Dbx1 neurons are necessary for spontaneous rhythmogenesis. However, whether these neurons are required under all conditions (e.g. alert adult mice) or whether rhythmogenesis can be evoked without these neurons is unclear.

Further, very little is known with regard to the role of non-Dbx1 glutamatergic neurons (or whether their role is easily distinguishable from Dbx1 neurons), although there is some evidence that they contribute more to "pattern" generation. Whether these specific neurons are necessary or sufficient for generating inspiratory bursts remains unknown (Since it is difficult to specifically target the non-Dbx1 glutamatergic population). Based on these considerations, we do not feel that it is a stretch to think that artificial stimulation of non-Dbx1 neurons could bypass the requirement for Dbx1 neurons to drive inspiration (Although this is unlikely the case under spontaneous conditions).

Thus, we agree with the reviewer that under spontaneous conditions the Dbx1 refractory period should be the rate limiting step (It is likely that these neurons initiate bursts because they are more excitable, and because they are more excitable they will have a longer refractory period). However, we do not think it is contradictory for artificial stimulation of Vglut2 to overcome the Dbx1 refractory period under evoked conditions.

2. Characterization of Dbx1 depolarization during expiration. In Figure 3F the authors activate Dbx1-derived neurons during expiration. According to their model, if Dbx1 neurons never recovery from the refractory period, then a burst should not occur. How is it that Dbx1 neurons recovery from the refractory period while they are depolarized by ChR2 during all of expiration. Whole cell recordings of Dbx1 neurons should be performed during the light pulse to characterize the extent of depolarization that occurs during a prolonged light pulse through expiration.

We agree with the reviewer, that if the neurons were not able to recover from the refractory period a subsequent burst should not occur. But we do not expect that activation of these neurons during expiration would “lock” them in a refractory mechanism. To the contrary, we suggest that stimulation during expiration accelerates burst onset following the refractory period by increasing the rate of spread of excitation through the network (percolation), whereas stimulation during inspiration exaggerates the refractory period and delays the onset of the next burst.

In Fig.2e we perform cell-specific current injections in isolated Dbx1 neurons at various times following an artificial “burst” to demonstrate how these neurons respond to depolarization during the refractory period. In addition, as the reviewer suggests, we now include an example trace of a whole cell recording of a Dbx1 neuron during light pulses, showing reduced excitation during the refractory period (Supplementary Fig. 2a).

The whole cell recordings in Figure 2e and supplemental Figure 2 suggest that the refractory period is an intrinsic property of the neurons which originates in the biophysical properties of their ion channels. The refractory period does not appear to be a property of the network as suggested above. Therefore, it seems that sustained depolarizing ChR2 stimulation during expiration, if sufficiently depolarizing, should maintain neurons in a state in which the voltage gated channels, like the sodium channels, stay desensitized. The interesting model of this paper and data supporting a refractory period requires a period of hyperpolarization to re-establish a state in which the next breath can be triggered. Thus, with the idea that ChR2 is depolarizing the neurons throughout expiration, it appears to contradict the model. In all likelihood, ChR2 itself is inactivating during this prolonged stimulation and therefore allowing for the hyperpolarized refractory period to take place. The only way to understand this would be to perform whole cell recordings of the Dbx1 neurons throughout the prolonged expiratory light stimulation. Although this may not be the most critical element of the paper, it is an important contradiction to a model implicating an cell autonomous refractory period defining expiratory length and should be addressed.

We show that preBotC neurons have intrinsic refractory mechanisms and that they likely contribute the refractory period at the population level. However, we do not rule out that synaptic interactions could also contribute to the refractory period as suggested by Kottick and Del Negro 2015.

We are not entirely clear what the reviewer is getting at with regard to ChR2 stimulation during the expiratory phase. What we observe is that targeted expiratory stimulation of Dbx1 or Vglut2 expressing neurons increases burst frequency, but that this increase is limited by the refractory period. Exactly how (at the level of ion channels) the refractory period limits the effects of ChR2 stimulation is not known and would be an interesting follow up study.

REVIEWERS' COMMENTS:

Reviewer #2 (Remarks to the Author):

The authors have responded appropriately to my critiques in their revisions. This is an important study that will have a lasting impact. I have a couple suggestions for the authors, which the editors can handle (I don't need to review them again).

I suggest to cut down on the use of superlatives, like "extremely" (line 58) and "dramatically" (line 139) because the reader can judge for herself how EXTREME and DRAMATIC the data are; the authors don't have to weigh in. In fact, the data are good and they will stand on their own without added emphasis.

line 219: are both statistics exactly the same, i.e., 10 +/- 3%?

Lines 355-357: Cregg et al. worked with in vitro brainstem-spinal cord preparations, not the "in situ" preparation which retains portions of the rostral body and ribcage of a rodent and then retrogradely perfuses the neuraxis via the aorta. Suggest changing "in situ" in line 357 to "in vitro".

Line 429, the references are garbled such that the citation is to "6278" as instead of, presumably, "62,78".

Reviewer #4 (Remarks to the Author):

Thank you for continuing to improve the clarity and certainty of the statements in the manuscript. In my view, the remaining points that I have raised are satisfactorily addressed and the manuscript should move to publication.

Regarding my comment about ChR2 activation during the entire duration of expiration. The point I am trying to raise is simple. For the glutamatergic neurons to become inactive, then the membrane potential of the cell must hyperpolarize. However, if you are instead artificially depolarizing the neuron with light, then you are preventing this step from occurring. Therefore, it is unclear to me how the neuron could recover in order to burst again and generate the next breath. The explanation is likely technical and would require more detailed recordings to understand the nature of the ontogenetic stimulation. At this time, I feel that it is fine for the manuscript to move forward to publication without this mystery being resolved.

REVIEWERS' COMMENTS:

Reviewer #2 (Remarks to the Author):

The authors have responded appropriately to my critiques in their revisions. This is an important study that will have a lasting impact. I have a couple suggestions for the authors, which the editors can handle (I don't need to review them again). ***We thank the reviewer for their large contribution to improving our manuscript***

I suggest to cut down on the use of superlatives, like "extremely" (line 58) and "dramatically" (line 139) because the reader can judge for herself how EXTREME and DRAMATIC the data are; the authors don't have to weigh in. In fact, the data are good and they will stand on their own without added emphasis. ***Corrected***

line 219: are both statistics exactly the same, i.e., 10 +/- 3%? ***This is correct***

Lines 355-357: Cregg et al. worked with in vitro brainstem-spinal cord preparations, not the "in situ" preparation which retains portions of the rostral body and ribcage of a rodent and then retrogradely perfuses the neuraxis via the aorta. Suggest changing "in situ" in line 357 to "in vitro". ***Corrected***

Line 429, the references are garbled such that the citation is to "6278" as instead of, presumably, "62,78". ***Corrected***

Reviewer #4 (Remarks to the Author):

Thank you for continuing to improve the clarity and certainty of the statements in the manuscript. In my view, the remaining points that I have raised are satisfactorily addressed and the manuscript should move to publication. ***We thank the reviewer for the efforts reviewing our manuscript***

Regarding my comment about ChR2 activation during the entire duration of expiration. The point I am trying to raise is simple. For the glutamatergic neurons to become inactive, then the membrane potential of the cell must hyperpolarize. However, if you are instead artificially depolarizing the neuron with light, then you are preventing this step from occurring. Therefore, it is unclear to me how the neuron could recover in order to burst again and generate the next breath. The explanation is likely technical and would require more detailed recordings to understand the nature of the ontogenetic stimulation. At this time, I feel that it is fine for the manuscript to move forward to publication without this mystery being resolved. ***We understand the reviewers point and agree that it would be interesting to study in detail how ChR2-mediated depolarization affects bursting properties of preBotC neurons. In our hands, it seems that preBotC neurons can continue to burst on top of a background ChR2-mediated inward current.***